

# Representation of population exchange at level anti-crossings

Bogdan A. Rodin and Konstantin L. Ivanov[*]

*International Tomography Center, Siberian Branch of the Russian Academy of Science, Novosibirsk, 630090, Russia;*

*Novosibirsk State University, Novosibirsk, 630090, Russia*

*\* Corresponding author, email: ivanov@tomo.nsc.ru*

**Abstract**

A theoretical framework is proposed to describe the spin dynamics driven by coherent spin mixing at Level Anti-Crossings (LACs). We briefly introduce the LAC concept and propose to describe the spin dynamics

using a vector of populations of the diabatic eigenstates. In this description, each LAC gives rise to a pairwise redistribution of eigenstate populations, allowing one to construct the total evolution operator of the spin system. Additionally, we take into account that in the course of spin evolution a "rotation" of the eigenstate basis case take place. The approach is illustrated by a number of examples, dealing with magnetic field inversion, cross-polarization, singlet-state NMR and parahydrogen induced polarization.



## 1 Introduction

NMR methods, which exploit coherent spin mixing at Level Anti-Crossings (LACs), are widely used in various areas of research, notably, to perform broad-band excitation (Baum et al., 1985; Freeman, 1998; Tannús and Garwood, 1997) and cross-polarization (Hartmann and Hahn, 1962), to transfer spin hyperpolarization (Ivanov et al., 2014; Theis et al., 2018; Theis et al., 2014b; Pravdivtsev et al., 2014c;

Pravdivtsev et al., 2014b; Pravdivtsev et al., 2014a; Franzoni et al., 2013) and to generate and detect long-lived nuclear singlet order (DeVience et al., 2013; Rodin et al., 2019; Rodin et al., 2018; Pravdivtsev et al., 2016). In this work, we propose an approach aimed at simple understanding of spin mixing at LACs and predicting the resulting spin order. The approach is applicable to spin systems with arbitrary populations of adiabatic nuclear spin states and no coherence between them; it makes use of two ingredients –

permutations of the populations and rotation of the basis of spin eigenstates. In this work, we introduce the main concept and formalism and provide a number of NMR-relevant examples, showing how the approach works. These examples include consideration of spin order transfer upon adiabatic inversion (Lukzen and Steiner, 1995; Eills et al., 2019) of the external magnetic field and, more generally, NMR experiments with field jumps (Miesel et al., 2006; Pravdivtsev et al., 2013a), as well as some pulsed NMR

experiments, such as cross-polarization (Hartmann and Hahn, 1962; Pines et al., 1972). Last but not least, using the language of LACs we describe some pulse sequences, which are currently exploited in singlet-state NMR (Levitt, 2019, 2012) and Para-Hydrogen Induced Polarization (PHIP) (Natterer and Bargon, 1997; Green et al., 2012; Barskiy et al., 2019; Duckett and Mewis, 2012).

PHIP makes use of the spin order of parahydrogen, $pH_2$, which is the $H_2$ molecule in its nuclear

singlet state. It is straightforward to enrich the $H_2$ gas in the *para*-component to $> 90\%$. Such a significant deviation of the singlet state population from the value expected at equilibrium conditions at high temperature, only 25% of $pH_2$, provides a source of strong non-thermal polarization. In the traditional PHIP method, $pH_2$ is attached to a substrate molecule by using a suitable catalyst. When the equivalence of the $pH_2$-nascent protons is broken in the reaction product, the non-thermal spin order can be converted

into observable magnetization, giving rise to significant NMR signal enhancements (Pravica and Weitekamp, 1988; Bowers and Weitekamp, 1987). PHIP can also be transferred from the primarily polarized protons to other nuclei in the product molecule to enhance their NMR signals. Alternatively, one can use the Signal Amplification By Reversible Exchange (SABRE) (Adams et al., 2009; Barskiy et al., 2019; Duckett and Mewis, 2012) method, in which no chemical modification of the substrate occurs.

Instead, $pH_2$ and the substrate bind to an Ir-based organometallic complex, where spin order conversion gives rise to polarization of the substrate. Subsequently, the hyperpolarized substrate molecule dissociates from the complex, contributing to polarization of the free substrate pool.

A related field is singlet-state NMR (Levitt, 2012; Carravetta and Levitt, 2004; Carravetta et al., 2004), dealing with slowly relaxing symmetry-protected spin states, which can be used to probe various

slow processes and to store non-equilibrium spin polarization. In many molecules (Levitt, 2012; Carravetta and Levitt, 2004; Carravetta et al., 2004; Stevanato et al., 2015; Sheberstov et al., 2019; Zhou et al., 2017; Wang et al., 2017; Buratto et al., 2014; Vasos et al., 2009; Zhang et al., 2015; Franzoni et al., 2012; Kiryutin et al., 2019; DeVience et al., 2013) singlet-order relaxes much longer than spin magnetization for the reason that it is immune to some relaxation mechanisms, for instance, in a two-spin system dipolar

relaxation cannot drive singlet-triplet transitions because the dipole-dipole interaction is invariant to exchange of the two spins (Pileio, 2010). In singlet-state NMR experiments, spin magnetization is converted into singlet order by a suitable pulse sequence; singlet-state readout is also done by singlet-to-magnetization conversion using special pulse sequences.

In the cases of PHIP and singlet-state NMR consideration of LACs often becomes important, in

particular, in molecules with pairs of nearly-equivalent spins (Ivanov et al., 2014; Pravdivtsev et al., 2013b; Franzoni et al., 2013; Franzoni et al., 2012; Sheberstov et al., 2019; Stevanato et al., 2015; DeVience et al.,



2013; Theis et al., 2014a), such that the symmetry breaking is due to a very small chemical shift difference of the nuclei or due to their magnetic non-equivalence, i.e., due to slightly different couplings to other spins. Such symmetry breaking can only occur under special conditions, which correspond to LACs. In this situation, the approach proposed in this work can be useful for understanding the spin dynamics.

This contribution aims at a simple description of LAC-based phenomena. We illustrate the concept presented here by a number of examples, in each case showing the scheme of energy levels and discussing the type of spin mixing. For numerical calculations, we used the "SpinDynamica" software package (Bengs and Levitt, 2018). We also anticipate that the present method is easy to exploit and widely applicable to treat magnetic resonance experiments, which utilize LACs.

## 2 Theory

### 2.1 Spin mixing at LACs

Before going into detail of the method, we would like to remind the reader the LAC concept (von Neumann and Wigner, 1929) and characterize the efficiency of spin mixing at LACs.

By a level anti-crossing, or an avoided crossing, we mean the following situation. Let us imagine a spin system described by the Hamiltonian

$$\widehat{\mathcal{H}} = \widehat{\mathcal{H}}_0 + \widehat{\mathcal{V}} \tag{1}$$

comprising the main term $\widehat{\mathcal{H}}_0$ and a small perturbation $\widehat{\mathcal{V}}$; we imply that the Frobenius norm of the perturbation term is much smaller, $\|\widehat{\mathcal{V}}\| \ll \|\widehat{\mathcal{H}}_0\|$. The perturbation term becomes relevant only under special conditions, namely, when the difference between energies dictated by the $\widehat{\mathcal{H}}_0$ term (eigenvalues of $\widehat{\mathcal{H}}_0$) is small, i.e., the energy levels tend to cross. Let us consider this situation in more detail.

Hereafter, we assume that there is a parameters $x$, which one can control experimentally: this can be the external magnetic field strength, or the strength of an applied radiofrequency (RF) field. Upon variation of $x$ the energies, i.e., eigenvalues of the spin Hamiltonian, change. For simplicity, we consider what happens to a pair of levels, corresponding to the eigenstates $|\psi_k\rangle$ and $|\psi_l\rangle$ of the "unperturbed" Hamiltonian $\widehat{\mathcal{H}}_0$, with energies $\mathcal{E}_k^0$ and $\mathcal{E}_l^0$, i.e., we consider the solutions of the eigenproblem $\widehat{\mathcal{H}}_0|\psi_{k,l}\rangle = \mathcal{E}_{k,l}^0|\psi_{k,l}\rangle$. The next step is to figure out how the perturbation term affects the actual energies and the corresponding eigenstates of the full Hamiltonian. When solving this problem, we assume that the energies $\mathcal{E}_k^0$ and $\mathcal{E}_l^0$ closely approach each other in a certain range of $x$ values, having a crossing at $x = x_0$ so that $\mathcal{E}_k^0(x_0) = \mathcal{E}_l^0(x_0)$. We also imply that all other states $|\psi_m\rangle$ (where $m \neq k, l$) are remote in energy at $x \approx x_0$. Below, we discuss the reason of making such an assumption. To solve the problem, we need to do nothing else but diagonalize the full Hamiltonian, including the perturbation term. To determine the actual state energies, i.e., the eigenvalues of $\widehat{\mathcal{H}}$, we solve the following equation for $\mathcal{E}$ and obtain the energies:

$$\begin{vmatrix} \mathcal{E} - \mathcal{E}_k^0 & \mathcal{V}_{kl} \\ \mathcal{V}_{lk} & \mathcal{E} - \mathcal{E}_l^0 \end{vmatrix} = 0 \quad \Rightarrow \quad \mathcal{E}_{k,l} = \frac{\mathcal{E}_k^0 + \mathcal{E}_l^0}{2} \pm \frac{1}{2}\sqrt{(\mathcal{E}_k^0 - \mathcal{E}_l^0)^2 + 4|\mathcal{V}_{kl}|^2} \tag{2}$$

For simplicity, here we assume that the perturbation term has only off-diagonal elements $\mathcal{V}_{kl} = \langle \psi_k|\widehat{\mathcal{V}}|\psi_l\rangle$ in the basis $|\psi_{k,l}\rangle$ (when this is not true the diagonal terms are also modified with a consequence that the actual crossing point might move from $x_0$ to $x_0'$). One can see that when $\mathcal{V}_{kl} \neq 0$, there are always two different solutions for the energy, $\mathcal{E}_k \neq \mathcal{E}_l$. Even when the unperturbed levels do cross, $\mathcal{E}_k^0(x_0) = \mathcal{E}_l^0(x_0)$, the levels of the total Hamiltonian are always different and cannot cross: the crossing is "avoided" and we obtain a LAC instead of the Level Crossing (LC), see **Figure 1a**. Or course, the perturbation term is inactive when $|\mathcal{V}_{kl}| \ll |\mathcal{E}_k^0 - \mathcal{E}_l^0|$ (since $\mathcal{E}_{k,l} \approx \mathcal{E}_{k,l}^0$) but it strongly affects the energies when $|\mathcal{V}_{kl}| \sim |\mathcal{E}_k^0 - \mathcal{E}_l^0|$. The range of $x$ values such that $|\mathcal{V}_{kl}| \sim |\mathcal{E}_k^0 - \mathcal{E}_l^0|$ determines the LAC



**MAGNETIC RESONANCE**
Open Access Discussions

region. The minimal splitting between $\mathcal{E}_k$ and $\mathcal{E}_l$ is achieved at the LC point $x_0$ (also giving the center of the LAC region) being equal to $2|\mathcal{V}_{kl}|$. According to the widely accepted terminology, the energy levels $\mathcal{E}_{k,l}^0$, corresponding to the unperturbed Hamiltonian, are diabatic levels, whereas the levels $\mathcal{E}_{k,l}$, corresponding to the full Hamiltonian, are adiabatic levels.

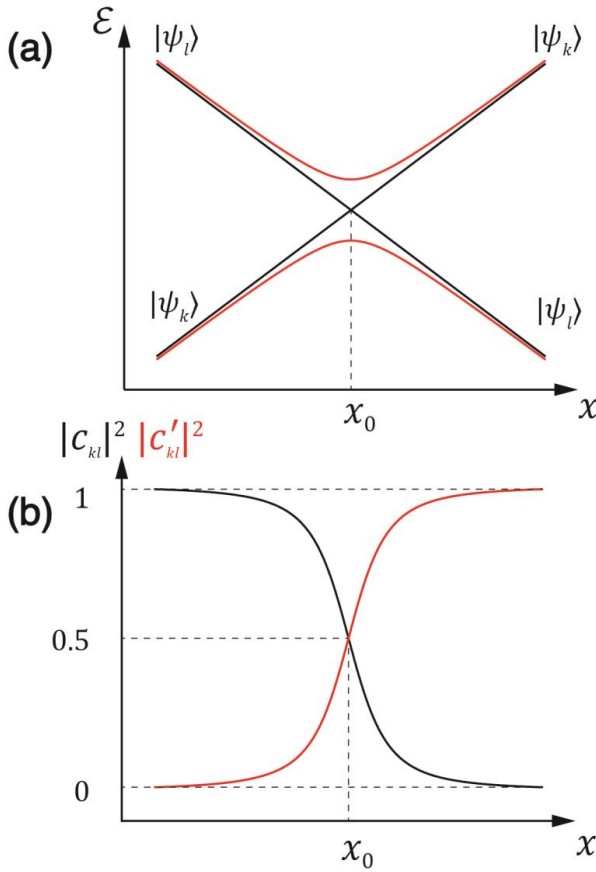

**Figure 1.** (a) Representation of an LC and LAC. The black lines represent the energies of the diabatic states, which have a LC. The red lines show the adiabatic energy levels. (b) The mixing coefficients introduced in eq. (3) in the LAC region.

It is important to emphasize that LACs strongly affect spin dynamics, giving rise to coherent spin mixing. To rationalize this, we need to solve the eigenproblem of the full Hamiltonian $\hat{\mathcal{H}}$. The two eigenstates corresponding to the levels $\mathcal{E}_k(x)$ and $\mathcal{E}_l(x)$ are superposition states of $|\psi_k\rangle$ and $|\psi_l\rangle$:

$$|\varphi_k\rangle = c_{kl}|\psi_k\rangle + c'_{kl}|\psi_l\rangle = \cos\theta_{kl}\,|\psi_k\rangle + \sin\theta_{kl}\,|\psi_l\rangle$$
$$|\varphi_l\rangle = -c'_{kl}|\psi_k\rangle + c_{kl}|\psi_l\rangle = -\sin\theta_{kl}\,|\psi_k\rangle + \cos\theta_{kl}\,|\psi_l\rangle \qquad (3)$$

The "mixing angle" $\theta_{kl}$ is defined via the off-diagonal perturbation term and the difference of the unperturbed energies:

$$\tan 2\theta = \frac{\mathcal{V}_{kl}}{\mathcal{E}_k^0 - \mathcal{E}_l^0} \qquad (4)$$

For the sake of simplicity, we assume that $\mathcal{V}_{kl}$ is real. The $\theta$ angle goes to zero when the unperturbed levels are very different in energy and the $|\psi_k\rangle$ and $|\psi_l\rangle$ states are the eigenstates of the spin system. However, in the LAC region $\theta \neq 0$ and the $|\psi_k\rangle$ and $|\psi_l\rangle$ states are superpositions of the true eigenstates $|\varphi_k\rangle$ and $|\varphi_l\rangle$. Hence, if initially the $|\psi_k\rangle$ state is populated, the spin system will not stay in this state: the





population will oscillate between the states $|\psi_k\rangle$ and $|\psi_l\rangle$. From eq. (3) we notice that this effect is particularly pronounced at $\mathcal{E}_k^0 - \mathcal{E}_l^0 = 0$, i.e., at the LC, $|\theta| = \frac{\pi}{4}$. In this case $|\varphi_{k,l}\rangle = \frac{1}{\sqrt{2}}\{|\psi_k\rangle \pm |\psi_l\rangle\}$

meaning that the population can be completely transferred between the states $|\psi_k\rangle$ and $|\psi_l\rangle$. This is exactly the way how LACs can be exploited: spin mixing at LACs can be utilized to perform complete transfer of the population from one state to another. In **Figure 1b** we demonstrate how the coefficients $c_{kl}$ and $c'_{kl}$, which describe the state mixing, change upon variation of the $x$ parameter: away from the LAC one of them goes to 1 and the other one goes to 0, whereas in the LAC region both of them are non-

zero. When a LC is not converted into a LAC mixing does not occur – for this reason, LCs are of no significance for this work.

Here we consider two different ways of transferring population between the diabatic states. The first method utilizes coherent spin mixing at the LAC. The idea is that away from the LAC we prepare the spin system in an unperturbed state, for clarity, in $|\psi_k\rangle$. A fast (non-adiabatic) jump to $x = x_0$ will keep

the state the same, but $|\psi_k\rangle$ now becomes a superposition of the true eigenstates

$$|\psi_k\rangle = \frac{1}{\sqrt{2}}\{|\varphi_k\rangle + |\varphi_l\rangle\} \tag{5}$$

The wavefunction will change in time, since the two eigenstates have different energies (having a LAC is equivalent to having two different energies). At time $t$, the wavefunction becomes (we express the energy in $\hbar$ units)

$$|\psi\rangle(t) = \frac{1}{\sqrt{2}}\{|\varphi_k\rangle e^{-i\mathcal{E}_k t} + |\varphi_l\rangle e^{-i\mathcal{E}_l t}\} \tag{6}$$

and the populations of the unperturbed state are (here we substitute $\mathcal{E}_k - \mathcal{E}_l = 2\mathcal{V}_{kl}$)

$$p_k = p(\psi_k) = |\langle \psi_k|\psi\rangle|^2 = \frac{1 + \cos(2\mathcal{V}_{kl}t)}{2}, \quad p_l = p(\psi_l) = |\langle \psi_l|\psi\rangle|^2 = \frac{1 - \cos(2\mathcal{V}_{kl}t)}{2} \tag{7}$$

Hence, the population oscillates between the states $|\psi_k\rangle$ and $|\psi_l\rangle$; at $t = \pi/2\mathcal{V}_{kl}$ the populations are inverted. If we bring the system out of the LAC at this instant of time the population will be transferred from $|\psi_k\rangle$ to $|\psi_l\rangle$. When $x \neq x_0$, coherent spin mixing can still take place but the efficiency of population exchange is reduced (e.g., population inversion is no longer possible).

Another possibility to transfer the population is to perform a slow (adiabatic) passage through the

LAC. When the adiabaticity condition is fulfilled, meaning that the rate of variation of $|\varphi_{k,l}\rangle$ is much smaller than the intrinsic evolution frequency of the spin system $|\mathcal{E}_k - \mathcal{E}_l|$, the populations adjust to the slow variation of the adiabatic eigenstates. As a consequence, the populations of the adiabatic eigenstates $|\varphi_{k,l}\rangle$ do not change upon passage through the LAC. This means that the populations of the diabatic states $|\psi_{k,l}\rangle$ are swapped: $p_k \to p_l$ and $p_l \to p_k$. Hence, like in the previous case, complete exchange of the

populations takes place. When complete adiabaticity is not achieved, the populations are not swapped, but partially redistributed, as explained in the following subsection. This effects can be taken into account by using the Landau-Zener approach (Zener, 1932). Specifically, assuming that initially $p_k = 1$ and $p_l = 0$, after a passage through a LAC we obtain the following state populations (Zener, 1932)

$$p_k = \exp\left[-\frac{2\pi|\mathcal{V}_{kl}|^2}{\mathcal{F}_{kl}}\right], \quad p_l = 1 - \exp\left[-\frac{2\pi|\mathcal{V}_{kl}|^2}{\mathcal{F}_{kl}}\right] \tag{8}$$

where $\mathcal{F}_{kl} = \frac{d}{dt}|\mathcal{E}_k - \mathcal{E}_l|$ gives the rate, at which the splitting between the diabatic levels changes in time

(in the Landau-Zener approach this speed is taken constant).

In many cases, adiabatic passage gives better results as compared to coherent exchange of populations, being more robust to inaccuracies in setting the parameters of the spin Hamiltonian. Indeed,



spin mixing using coherences requires that $x$ is precisely set to satisfy the LC condition for the Hamiltonian $\hat{\mathcal{H}}_0$ and the timing is controlled. In the case of adiabatic passage, it is sufficient to pass through the LAC region slowly enough. One should note, however, that as far as the transfer time is concerned, coherent population exchange is preferable, since it takes less time (an adiabatic process always requires a relatively slow variation of the control parameter).

We illustrate how population exchange can take place for spin-$\frac{1}{2}$, i.e., in a two-level system, which is described by the Hamiltonian:

$$\hat{\mathcal{H}}(t) = \hat{\mathcal{H}}_0(t) + \hat{\mathcal{V}}, \quad \hat{\mathcal{H}}_0(t) = \omega_z(t)\hat{I}_z, \quad \hat{\mathcal{V}} = \omega_x\hat{I}_x \qquad (9)$$

Hence, a time-dependent field is applied along the $z$-axis; additionally there is a constant $x$-field. The system has a LAC at zero magnetic field, where the $|\alpha\rangle$ and $|\beta\rangle$ eigenstates of $\hat{\mathcal{H}}_0$ have a crossing, which is "avoided" due to the presence of the perturbation term. As usual, by $|\alpha\rangle$ and $|\beta\rangle$ we hereafter denote the spin-1/2 states with the $z$-projection of $+\frac{1}{2}$ and $-\frac{1}{2}$, respectively.

If we assume that initially the system is in the $|\alpha\rangle$ state, a possible way to perform the $|\alpha\rangle \rightarrow |\beta\rangle$ population transfer is to introduce a non-adiabatic jump to zero field, where the true eigenstates, $(|\alpha\rangle \pm |\beta\rangle)/\sqrt{2}$ are superposition states of $|\alpha\rangle$ and $|\beta\rangle$. In this situation, according to eq. (7), the population oscillates between $|\alpha\rangle$ and $|\beta\rangle$, as shown in **Figure 2a**.

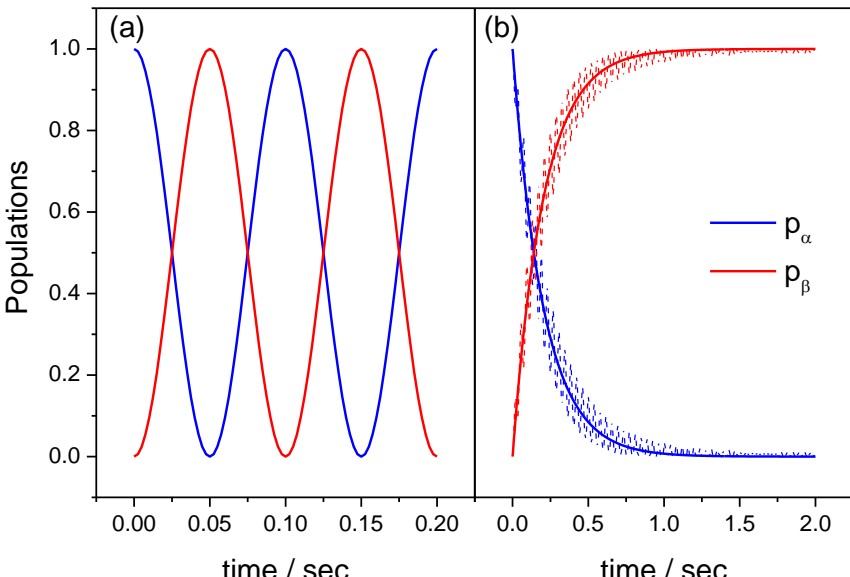

**Figure 2.** (a) Time dependence of the state populations in the case of coherent exchange with $\omega_x/2\pi = 10$ Hz, as obtained from eq. (7). (b) Populations after a passage through zero field, $\omega_z = 0$, as functions of the switching time $\tau_{sw}$. Here the solid lines present the result of eq. (8) and the dashed lines show the numerical simulation result. Here $\omega_x/2\pi = 10$ Hz, $\omega_z^{max}/2\pi = 100$ Hz. Initially the system is in the $|\alpha\rangle$ state; the blue and red lines shown the populations of the $|\alpha\rangle$ and $|\beta\rangle$ states, respectively.

Another possibility is to perform an adiabatic passage through the LAC, by varying the $z$-component of the field, so that $\omega_z$ goes from a negative value $-\omega_{max}$ to a positive value $+\omega_{max}$. Here we assume that $\omega_{max} \gg \omega_x$ and that the time dependence of $\omega_z$ is a linear dependence

$$\omega_z(t) = \omega_{max}\left(\frac{2t}{\tau_{sw}} - 1\right)$$



with $\tau_{sw}$ being the duration of the switch. The resulting state populations would then follow from eq. (8), with the Landau-Zener parameter equal to

$$\frac{2\pi|\mathcal{V}_{kl}|^2}{\mathcal{F}_{kl}} = \frac{\pi\omega_x^2\tau_{sw}}{4\omega_z^{max}}$$

The resulting state populations are shown in **Figure 2b**; for comparison we also show the result of a numerical simulation of the spin dynamics with a $\widehat{\mathcal{H}}(t)$ time-dependent Hamiltonian.

We would like to emphasize that in some cases the mixing matrix element is zero; however, when the states $|\psi_k\rangle$ and $|\psi_l\rangle$ are both coupled to a third state $|\psi_m\rangle$ the basis wavefunctions also become perturbed and a mixing matrix element $\mathcal{V}_{kl}$ effectively becomes non-zero. In **Appendix A**, we explain how

to calculate $\mathcal{V}_{kl}$ in this case, corresponding to degenerate perturbation theory. Hence, the two states $|\psi_k\rangle$ and $|\psi_l\rangle$ are never mixed (and the LC is never converted to a LAC) only when the Hamiltonian $\widehat{\mathcal{H}}$ is block-diagonal and these two states belong to different blocks.

### 2.2 Theoretical framework

The idea of this paper is to describe how spin order changes due to coherent spin mixing at LACs.

In all cases, we consider processes, in which a certain parameter $x(t)$ is varied so that the spin Hamiltonian $\widehat{\mathcal{H}}_0(x)$ also varies with time and the system goes through LCs, which are converted into LACs by the $\widehat{\mathcal{V}}$ term. In the following, we make several assumptions.

First, we consider the initial and final spin states characterized by the density matrices $\rho_i$ and $\rho_f$, which are diagonal in the eigenbasis of the Hamiltonian:

$$\rho_i = \sum_m p_m |\psi_m^i\rangle\langle\psi_m^i|, \quad \rho_f = \sum_n p_n |\psi_n^f\rangle\langle\psi_n^f| \tag{10}$$

where $|\psi_m^i\rangle$ and $|\psi_m^f\rangle$ stand for the diabatic eigenstates of the initial and final unperturbed Hamiltonian $\widehat{\mathcal{H}}_0$. We also assume that the eigenstates of $\widehat{\mathcal{H}}_0$ can be determined analytically at any $x$ value, which is possible in many cases when the perturbation term is dropped off. Consideration of the coherences can be complicated, as they give rise to complex phenomena, e.g., those described by the Berry phase (Zwanziger et al., 1990; Berry, 1984). Here we avoid such complexities assuming that the initial state is

adjusted such that the density matrix $\rho_i$ is diagonal in the eigenbasis of the initial Hamiltonian. This means that instead of the density matrix we can use a vector of state populations, $|\rho\rangle$, introduced in the following way:

$$|\rho\rangle = \sum_m p_m |\psi_m\rangle \tag{11}$$

Here $|\psi_m\rangle = |\psi_m\rangle\langle\psi_m|$ define the operator basis for the density matrix. It is easy to notice, that this basis is orthonormal as $(\psi_m|\psi_k) \equiv \mathrm{Tr}\{|\psi_m\rangle\langle\psi_m|\psi_k\rangle\langle\psi_k|\} = \delta_{mk}$. Since we deal with population vectors, in

eq. (10) we omit all terms $|\psi_m\rangle\langle\psi_n|$ when $m \neq n$.

Second, we assume that the spin dynamics is described entirely in terms of redistribution of the populations, occurring at LACs. The idea is that we can determine the LC points for the levels of $\widehat{\mathcal{H}}_0$, figure out whether the LCs are converted into LACs by the $\widehat{\mathcal{V}}$ term and assume that at each LAC redistribution of the corresponding state populations is taking place. This means that after mixing at the LAC between $|\psi_k\rangle$

and $|\psi_l\rangle$ the populations of the diabatic eigenstates change as follows

$$p_k \to p_k' = (1 - \Delta_{kl})p_k + \Delta_{kl}p_l$$
$$p_l \to p_l' = (1 - \Delta_{kl})p_l + \Delta_{kl}p_k \tag{12}$$





Here $\Delta_{kl}$ stands for the mixing efficiency, which is varied between zero and 1. Hence, we keep in mind that exchange of the populations may be incomplete, for instance, when the time of the coherent evolution at the LAC is not optimized or when the adiabaticity condition is not perfectly fulfilled. When $\Delta_{kl} = 1$ the populations are swapped, when $\Delta_{kl} = 0$ there is no population exchange taking place. The

precise $\Delta_{kl}$ value can be determined by simulating the spin dynamics at the LAC. For coherent spin mixing and adiabatic passage it can be determined from eqs. (7) and (8), respectively.

Third, we assume that LACs are isolated from each other, meaning that the spin mixing is occurring independently at different LACs. For instance, the region of LAC occurring between the states $|\psi_k\rangle$ and $|\psi_l\rangle$ should not overlap with that of the LAC between the states $|\psi_k\rangle$ and $|\psi_m\rangle$. LACs between different

pairs of states are allowed to occur at similar values of $x$. Under such assumptions we can describe the spin dynamics in terms of pairwise redistribution of populations at isolated LACs.

Finally, we need to consider that the eigenstates of the Hamiltonian $\hat{\mathcal{H}}_0$ can differ when the $x$ parameter is varied: a "rotation" of the eigenbasis can take place. The state basis $|\psi_m^f\rangle$ is then "tilted" with respect to the basis $|\psi_m^i\rangle$. Hence, when we compute an expectation value of a certain spin operator

in the basis of $|\psi_m^f\rangle$ states, it might correspond to a different operator in the $|\psi_m^i\rangle$ basis. This happens, for instance, when the direction of a quantization axis changes upon variation of $x$. We will discuss such examples separately.

Using these assumptions, we can formulate the theory for evaluating the spin evolution driven by LACs. Redistribution of the diabatic state populations given by eq. (12) can be described by an operator

$\hat{\Pi}^{(kl)}(\Delta_{kl})$, hereafter, termed as "population redistribution operator", which is a square matrix with the following non-zero elements (here $\delta_{mn}$ is the Kronecker delta):

$$\hat{\Pi}_{mn}^{(k,l)}(\Delta_{kl}) = \delta_{mn} \text{ (when } m, n \neq k, l) \tag{13}$$
$$\hat{\Pi}_{kk}^{(k,l)}(\Delta_{kl}) = \hat{\Pi}_{ll}^{(k,l)}(\Delta_{kl}) = 1 - \Delta, \quad \hat{\Pi}_{kl}^{(k,l)} = \hat{\Pi}_{lk}^{(kl)} = \Delta$$

This operator can be explicitly written as:

$$\hat{\Pi}_{mn}^{(kl)} = (1-\Delta)\{|\psi_k^f\rangle\langle\psi_k^i| + |\psi_l^f\rangle\langle\psi_l^i|\} + \Delta\{|\psi_l^f\rangle\langle\psi_k^i| + |\psi_k^f\rangle\langle\psi_l^i|\} + \sum_{m\neq l,m}|\psi_m^f\rangle\langle\psi_m^i| \tag{14}$$

Acting on the vector of populations by $\hat{\Pi}^{(kl)}$ we get the result

$$|\rho'\rangle = \hat{\Pi}^{(k,l)}(\Delta_{kl})|\rho\rangle \tag{15}$$

The elements of the new population vector $|\rho'\rangle$ are $p_j' = p_j$ for $j \neq k, l$; the $p_k'$ and $p_l'$ populations are

given by eq. (12). To be more precise, one should term $\hat{\Pi}$ "superoperator" (as it is an operator acting in the operator space); however, we do not use double "hats" and omit this complexity for the sake of brevity.

If the system passes through a sequence of LACs (occurring in pairs of state $kl, \dots, pq, rs$) the resulting redistribution operator is

$$\hat{\Pi} = \hat{\Pi}^{(rs)}(\Delta_{rs}) \cdot \hat{\Pi}^{(pq)}(\Delta_{pq}) \cdot \dots \cdot \hat{\Pi}^{(kl)}(\Delta_{kl}) \quad \Rightarrow \quad |\rho'\rangle = \hat{\Pi}|\rho\rangle \tag{16}$$

The operators, describing population redistribution at subsequent LACs, are sequentially multiplied from right to left to obtain the resulting operator $\hat{\Pi}$.

In some cases, the actual permutation of the state populations is performed via several consecutive permutations, for example, $i \to p \to f$. Such a sequence of simple permutations gives rise to a more complex permutation. When $\Delta = 1$ for each permutation, the actual form of the $\hat{\Pi}$ operator is

simplified, corresponding to cyclic permutation. For instance, for permutations $i \to p \to f$ we obtain $\hat{\Pi} = \hat{\Pi}^{(pf)}(1) \cdot \hat{\Pi}^{(ip)}(1)$. This is equivalent to the following permutations: $i \to f, f \to p, p \to i$. In this work we





will mostly consider spin order transfer pathways with a single permutation. Nevertheless, we also discuss cases where more complex permutations come into play (Rodin et al., 2020).

Knowing the final vector or state populations, we are able to evaluate the final density matrix from eq. (15) and to compute the expectation values of a spin operator $\hat{Q}_A$ of interest:

$$Q_A = (\hat{Q}_A | \rho') = \text{Tr}\{Q_A \cdot \rho'\} \tag{17}$$

It is important to note that that for many operators the $Q_A$ expectation value will be zero, because all off-diagonal elements of the density matrix are zero. In some cases, it is desirable to express the resulting spin order in the eigenbasis $|\psi_k^i\rangle$ of the initial Hamiltonian: an additional transformation is required then described by a basis rotation operator $\hat{\Psi}_{i \to f}$ (whereas $\hat{\Psi}_{f \to i}$ gives the inverse transformation). If we then
express the final density matrix in the initial $|\psi_k^i\rangle$ basis, it becomes as follows:

$$|\rho) = \sum_m p_m \, \hat{\Psi}_{f \to i} |\psi_m) \tag{18}$$

In some cases, the basis rotation is equivalent to a physical rotation of spins in the three-dimensional space. In this situation, we can introduce the rotation axis $\mathbf{n}$ and the rotation angle $\vartheta$, so that $\hat{\Psi}_{i \to f} = \hat{\Psi}_{\mathbf{n}}(\vartheta)$, where $\hat{\Psi}_{\mathbf{n}}(\vartheta)$ is the superoperator describing the actual rotation (all superoperators here are denoted by capital Greek letters). Using expression (18), we can evaluate the expectation value of any
operator of interest.

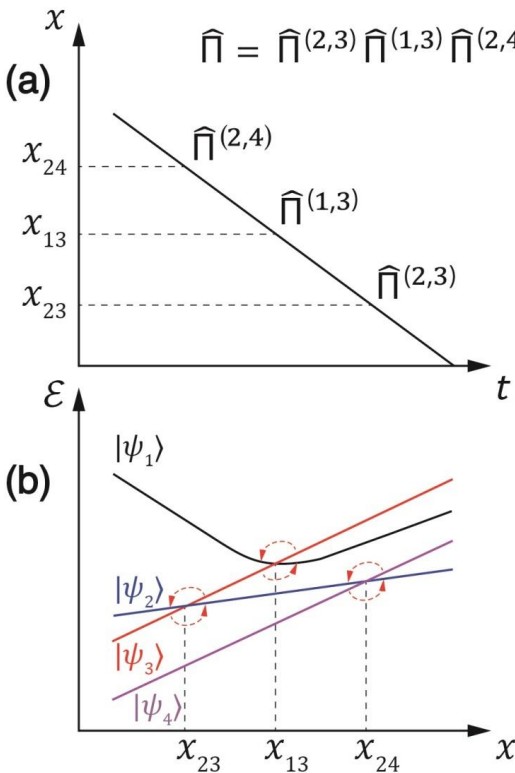

**Figure 3.** (a) Variation of the control parameter $x$ with time. When the $x$ value reaches the LACs region, mixing of the populations occurs, LC positions $x_{kl}$ are indicated as well as the permutation operators. (b) Representation of the population mixing between the pairs of diabatic states at the corresponding LACs, indicated by arrows.





265       Basis "rotation" becomes an important concern in some NMR experiments: an example is given by our recent work (Rodin et al., 2020) on "algorithmic cooling" of a spin system exploiting long-lived singlet order. The protocol for algorithmic cooling requires specific permutations of state populations in a four-level system, which are carried out by using NMR pulses with adiabatically increased or decreased field strength (which make use of adiabatic passage through LACs). Such pulses not only swap state

populations but also rotate the basis of spin eigenstates. Consequently, additional pulses are required to compensate for this effect (Rodin et al., 2020). Examples, in which basis rotation is taking place, are discussed below.

      The conversion of spin order can be illustrated by a diagram, as the one depicted in **Figure 3**. In the diagram above, we plot the $x(t)$ trajectory in a schematic way, showing only the passages through

LACs or jumps to LACs. In the diagram below, we show the energy levels as functions of $x$ and indicate the pathway for redistribution of the state populations. The resulting spin order can be represented by the populations of the eigenstates $|\psi_n^f\rangle$ in the cases of either complete population exchange or partial redistribution of the populations.

## 3 Results and Discussion

280       In this section, we consider a number of examples of LC/LAC based analysis of the spin dynamics. In each case, we start from introducing the $\widehat{\mathcal{H}}_0$ Hamiltonian (along with its eigenvalues) and the perturbation term $\widehat{V}$. After that, we explain how spin order of the system is modified due to the evolution at LACs.

### 3.1 Adiabatic zero-field passage

285       The first example we consider here is given by adiabatic inversion of the external magnetic field $\mathbf{B}||z$. The simplest example is given by a two-spin system with spins $I$ and $S$ of different kind, i.e., two heteronuclei with the gyromagnetic ratios $\gamma_I \neq \gamma_S$.

      The Hamiltonian of the spin system is given by expression (in $\hbar$ units; here $J_{IS}$ is the coupling strength, given in Hz)

$$\widehat{\mathcal{H}} = -\gamma_I B \hat{I}_z - \gamma_S B \hat{S}_z + 2\pi J_{IS}(\hat{\mathbf{I}} \cdot \hat{\mathbf{S}}) \tag{19}$$

Here we assume that the first two terms and the secular part of the coupling term give the main Hamiltonian

$$\widehat{\mathcal{H}}_0 = -\gamma_I B \hat{I}_z - \gamma_S B \hat{S}_z + 2\pi J_{IS} \hat{I}_z \hat{S}_z$$

while the non-secular coupling term is a perturbation:

$$\widehat{V} = \pi J_{IS}\{\hat{I}_+\hat{S}_- + \hat{I}_-\hat{S}_+\}$$

The unperturbed states of the spin system are the Zeeman states $|1\rangle = |\alpha\alpha\rangle$, $|2\rangle = |\alpha\beta\rangle$, $|3\rangle = |\beta\alpha\rangle$ and $|4\rangle = |\beta\beta\rangle$.

      When $B = 0$ the unperturbed energy levels cross: $|\alpha\alpha\rangle$ crosses with $|\beta\beta\rangle$ and $|\alpha\beta\rangle$ crosses with $|\beta\alpha\rangle$. The first LC cannot be converted into a LAC, since $\langle\alpha\alpha|\widehat{V}|\beta\beta\rangle = 0$, but the second LC is turned into a LAC by the perturbation term, since $\langle\alpha\beta|\widehat{V}|\beta\alpha\rangle = \pi J_{IS}$. The true LCs are completely irrelevant for spin

mixing, but at the LAC the populations of the states $|2\rangle$ and $|3\rangle$ can be exchanged. The energy levels are schematically shown in **Figure 4**. One can see that there are two more LCs at $B \neq 0$ (an LC at a positive field and an LC at a negative field), which are never converted to LACs, because the corresponding states are characterized by different values of the $z$-projection of the total spin $\hat{\mathbf{F}} = \hat{\mathbf{I}} + \hat{\mathbf{S}}$ and are not mixed by



the perturbation term. However, mixing at this LCs may become a concern (Lukzen and Steiner, 1995) in
the presence of an additional transverse field. Discussing such effects is beyond the scope of this work.

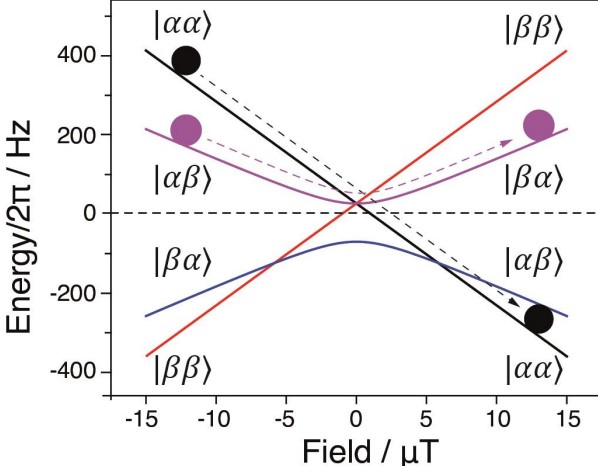

**Figure 4.** Correlation diagram for an adiabatic inversion. Simulation parameters: $J_{IS} = 100$ Hz, $I$ and $S$ spins are $^1$H and $^{13}$C nuclei with the gyromagnetic ratios $\gamma_H = 2.68 \cdot 10^8$ rad·s$^{-1}$·T$^{-1}$ and $\gamma_C = 6.73 \cdot 10^7$ rad·s$^{-1}$·T$^{-1}$.

If we assume that the two spins have different polarizations, the initial density matrix is given by
expression

$$\rho = \frac{1}{4}\hat{1} + M_I\hat{I}_z + M_S\hat{S}_z \qquad (20)$$

and the population vector in the basis of Zeeman states, $\mathbb{Z} = \{\alpha\alpha, \alpha\beta, \beta\alpha, \beta\beta\}$, is as follows:

$$|\rho) = \begin{pmatrix} \frac{1}{4} + \frac{1}{2}M_I + \frac{1}{2}M_S \\ \frac{1}{4} + \frac{1}{2}M_I - \frac{1}{2}M_S \\ \frac{1}{4} - \frac{1}{2}M_I + \frac{1}{2}M_S \\ \frac{1}{4} - \frac{1}{2}M_I - \frac{1}{2}M_S \end{pmatrix} \qquad (21)$$

The coefficients $M_I = \text{Tr}\{\hat{I}_z\rho\}$ and $M_S = \text{Tr}\{\hat{S}_z\rho\}$ give the polarizations of the two nuclei, which are taken different, $M_I \neq M_S$.

Now we consider a passage through zero field from $-B_0$ to $+B_0$, assuming that $|\gamma_I - \gamma_S|B_0 \gg$
$2\pi|J_{IS}|$ (this condition simply means that at $B = \pm B_0$ the spin system is away from the LAC region). If we redistribute the populations of the states $|2\rangle$ and $|3\rangle$ by an adiabatic passage through the LAC, we arrive at the following expression for the populations

$$|\rho') = \hat{\Pi}^{(\alpha\beta,\beta\alpha)}(\Delta)|\rho) = \begin{pmatrix} \frac{1}{4} + \frac{1}{2}M_I + \frac{1}{2}M_S \\ \frac{1}{4} + (1 - 2\Delta)\left[\frac{1}{2}M_I - \frac{1}{2}M_S\right] \\ \frac{1}{4} - (1 - 2\Delta)\left[\frac{1}{2}M_I - \frac{1}{2}M_S\right] \\ \frac{1}{4} - \frac{1}{2}M_I - \frac{1}{2}M_S \end{pmatrix} \qquad (22)$$

Rewriting the $\hat{I}_z$ and $\hat{S}_z$ operators in their vector form (i.e., omitting zero off-diagonal elements):



$$(I_z| = \begin{pmatrix} \dfrac{1}{2} & \dfrac{1}{2} & -\dfrac{1}{2} & -\dfrac{1}{2} \end{pmatrix}$$
$$(S_z| = \begin{pmatrix} \dfrac{1}{2} & -\dfrac{1}{2} & \dfrac{1}{2} & -\dfrac{1}{2} \end{pmatrix}$$

(23)

we can determine the polarization values:

$$M_I' = (I_z|\rho') = (1-\Delta)M_I + \Delta M_S, \quad M_S' = (S_z|\rho') = (1-\Delta)M_S + \Delta M_I$$

(24)

Hence, redistribution of polarizations occurs. When the efficiency $\Delta = 1$ we obtain that the spins exchange polarizations, $M_I' = M_S$ and $M_S' = M_I$, in accordance with an earlier result on polarization transfer in electron-nuclear systems (Lukzen and Steiner, 1995).

Polarization transfer can be carried out in other ways. For instance, one can perform a non-adiabatic jump $B_0 \to B = 0$, i.e., to the LAC, to convert the population difference $(p_{\alpha\beta} - p_{\beta\alpha})$ into the

coherences between the new eigenstates $|2,3\rangle = \{|\alpha\beta\rangle \pm |\beta\alpha\rangle\}/\sqrt{2}$. As explained above, by controlling the evolution time at zero-field one can change the sign of the coherence. After that, a non-adiabatic field jump to $B_0$ will swap the populations of the states $|2\rangle$ and $|3\rangle$. If we assume that the mixing efficiency $\Delta$ is less than one, we get the general result given by eq. (24).

In this context, it is useful to consider a more complex problem of enhancing NMR signals of

"insensitive" nuclei, such as $^{13}C$ or $^{15}N$, by transferring PHIP upon adiabatic passage through zero field. This method has been successfully implemented (Eills et al., 2019) to polarize $^{13}C$ nuclei in a system of two protons prepared in the singlet spin state and a carbon nucleus. In this case, two protons (spins $I_a$ and $I_b$) coupled to a $^{13}C$ nucleus (spin $S$), the spin Hamiltonian takes the form:

$$\widehat{\mathcal{H}} = -\gamma_I B\{\hat{I}_{az} + \hat{I}_{bz}\} - \gamma_S B\hat{S}_z + 2\pi J_{HH}(\hat{\mathbf{I}}_a \cdot \hat{\mathbf{I}}_b) + 2\pi J_{aS}(\hat{\mathbf{I}}_a \cdot \hat{\mathbf{S}}) + 2\pi J_{bS}(\hat{\mathbf{I}}_b \cdot \hat{\mathbf{S}})$$

(25)

The proton-proton coupling is $J_{HH}$; the coupling on the first proton and second proton to the carbon

nucleus are denoted as $J_{aS}$ and $J_{bS}$. The key issue is how to separate the Hamiltonian into two parts. Hereafter, we follow the results of Eills et al. (Eills et al., 2019) introducing the main Hamiltonian as (keeping Zeeman interactions, proton-proton coupling and secular part of the heteronuclear couplings)

$$\widehat{\mathcal{H}}_0 = -\gamma_I B\{\hat{I}_{az} + \hat{I}_{bz}\} - \gamma_S B\hat{S}_z + 2\pi J_{HH}(\hat{\mathbf{I}}_a \cdot \hat{\mathbf{I}}_b) + 2\pi J_{aS}\hat{I}_{az}\hat{S}_z + 2\pi J_{bS}\hat{I}_{bz}\hat{S}_z$$

and the perturbation as

$$\hat{V} = \pi J_{aS}\{\hat{I}_{a+}\hat{S}_- + \hat{I}_{a-}\hat{S}_+\} + \pi J_{bS}\{\hat{I}_{b+}\hat{S}_- + \hat{I}_{b-}\hat{S}_+\}$$

For $\widehat{\mathcal{H}}_0$ the eigen-basis of states is the "singlet-triplet-Zeeman" basis. In ref. (Eills et al., 2019) such a basis is introduced in two ways. The obvious one is to use the basis $\mathbb{STZ} = \{|S\rangle, |T_+\rangle, |T_0\rangle, |T_-\rangle\}_{12} \otimes \{|\alpha\rangle, |\beta\rangle\}_S$. This is the singlet-triplet basis of the $I$ spins and Zeeman basis of the $S$ spin. As usual, the singlet-triplet states are

$$|S\rangle = \frac{|\alpha\beta\rangle - |\beta\alpha\rangle}{\sqrt{2}}, \quad |T_+\rangle = |\alpha\alpha\rangle, \quad |T_0\rangle = \frac{|\alpha\beta\rangle + |\beta\alpha\rangle}{\sqrt{2}}, \quad |T_-\rangle$$

(26)

However, one should note that the true eigenbasis of $\widehat{\mathcal{H}}_0$ is given by $\mathbb{STZ}' \neq \mathbb{STZ}$, which takes into account that the only the states $|T_\pm\alpha\rangle$ and $|T_\pm\beta\rangle$ are true eigenstates of $\widehat{\mathcal{H}}_0$, while the other four states are superposition states of $|S\alpha\rangle$, $|S\beta\rangle$, $|T_0\alpha\rangle$ and $|T_0\beta\rangle$. However, when $J_{HH}$ is significantly larger than the other two couplings in the spin system, the following expressions hold approximately: $|S\alpha\rangle' \approx |S\alpha\rangle$, $|S\beta\rangle' \approx |S\beta\rangle$, $|T_0\alpha\rangle' \approx |T_0\alpha\rangle$ and $|T_0\beta\rangle' \approx |T_0\beta\rangle$. In this situation, assuming a special case of the spin

system prepared in the $|S\rangle$ state of the $I$ spins, we can approximately set only four populations to a non-zero value, namely, the populations of the $|S\alpha\rangle'$, $|S\beta\rangle'$, $|T_0\alpha\rangle'$ and $|T_0\beta\rangle'$.





In the spin system, there is a number of LCs and LACs, see **Figure 5**. At zero-field, in any multi-spin system there are always several LCs present (for symmetry reasons, groups of spin states become degenerate): in the present case six levels with proton triplet character are degenerate, as well as the two states having singlet character. There is also a number of LCs at non-zero fields, however, not all of them are turned into LACs. The reason is the same as in the case of an $IS$ two-spin system: all terms in $\hat{\mathcal{H}}$ do not alter the $z$-projection of all three spins, $\hat{\mathbf{F}} = \hat{\mathbf{I}}_a + \hat{\mathbf{I}}_b + \hat{\mathbf{S}}$. For this reason, we need to consider only four LCs, which turn into LACs. The LC positions have been determined in the previous work (Eills et al., 2019); they are as follows, $B_{LC}^{(1)}$ and $B_{LC}^{(2)}$:

$$B_{LC}^{(1)} = \frac{\pi}{2} \cdot \frac{4J_{HH} - J_\Sigma}{\gamma_I - \gamma_S}, \qquad B_{LC}^{(2)} = -\frac{\pi}{2} \cdot \frac{J_\Sigma}{\gamma_I - \gamma_S} \qquad (27)$$

where $J_\Sigma = J_{1S} + J_{2S}$; this expression is valid when $|J_{HH}| \gg |J_{1S} - J_{2S}|$. Upon adiabatic passage $-B_0 \rightarrow +B_0$ (where $B_0 \gg B_{LC}^{(1)}, B_{LC}^{(2)}$) the following population swapping occurs

$$|S\alpha\rangle' \leftrightarrow |T_+\beta\rangle'$$
$$|S\beta\rangle' \leftrightarrow |T_-\alpha\rangle' \leftrightarrow |T_0\beta\rangle' \equiv |S\beta\rangle' \leftrightarrow |T_0\beta\rangle' \qquad (28)$$

Strictly speaking, upon the field inversion two more population swaps occur: additionally there are population swaps of the kind $|T_0\beta\rangle' \leftrightarrow |T_-\alpha\rangle'$ and $|T_+\beta\rangle' \leftrightarrow |T_0\alpha\rangle'$. Hence, in both state manifolds with $F_z = \pm\frac{1}{2}$, we have cyclic permutations of the populations of three states. However, initially only one of the three states of each manifold (the one with singlet character of the protons) is populated, which simplifies the description. Specifically, in the $F_z = +\frac{1}{2}$ manifold it is sufficient to consider a single population swap, whereas in the $F_z = -\frac{1}{2}$ manifold two swaps should be taken into account.

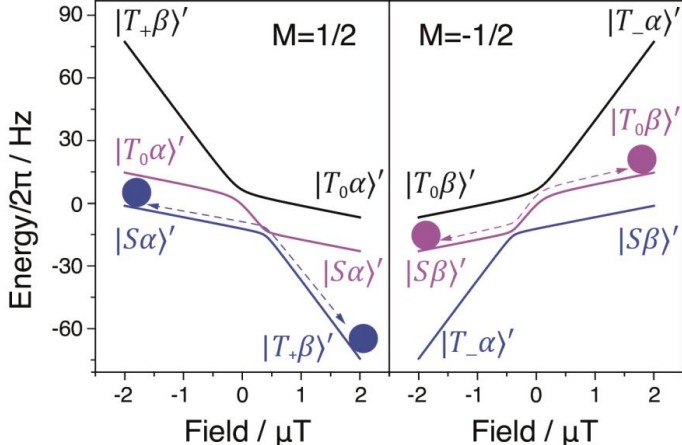

**Figure 5.** The two state manifolds of a three-spin $I_1I_2S$ system with $F_z = \pm 1/2$. The balls represent the state populations in the initial and final state, while the arrows show the adiabatic pathways. Simulation parameters: $I$ and $S$ nuclei are $^1$H and $^{13}$C respectively, $J_{HH} = 15.7$ Hz, $J_{1S} = 6.6$ Hz, $J_{2S} = 3.2$ Hz.

The initial density matrix in the case under study can be written as:

$$|\rho\rangle \approx \frac{1}{2}|S\alpha\rangle' + \frac{1}{2}|S\beta\rangle' \qquad (29)$$

After the adiabatic swap, the final density matrix becomes (when $\Delta = 1$ for the relevant LACs)

$$|\rho'\rangle = \frac{1}{2}\hat{\Pi}^{(T_+\beta,S\alpha)}|S\alpha\rangle' + \frac{1}{2}\hat{\Pi}^{(T_0\beta,S\beta)}|S\beta\rangle' = \frac{1}{2}|T_+\beta\rangle' + \frac{1}{2}|T_0\beta\rangle' \qquad (30)$$



As a result, the singlet order is converted into $z$-polarization of protons and $S$ spins. The polarizations of the $I$ spins and $S$ spins becomes (if we assume that only two states are populated at $B = B_0$)

$$M_S = (S_z|\rho') = \frac{1}{2}[(S_z|T_+\beta)' + (S_z|T_0\beta)'] = -\frac{1}{2} \tag{31}$$

$$M_I = (I_z|\rho') = \frac{1}{2}[(I_z|T_+\beta)' + (I_z|T_0\beta)'] = \frac{1}{2}$$

Hence, the singlet order is converted into the polarization of the $I$ spins and $S$ spins; $M_I$ and $M_S$ are the same in size but have opposite signs, since the $F_z$ value is conserved.

Spin order transfer in this system can be carried out in a different (perhaps, simpler) way. For instance, one can perform a sweep from $B = 0$ to $+B_0$: the populations are swapped between the states $|S\alpha\rangle' \leftrightarrow |T_+\beta\rangle'$ whereas the population of the $|S\beta\rangle'$ state remain the same. One more possibility is to perform a non-adiabatic field jump $B_0 \rightarrow B_{LC}^{(1)}$ to generate the coherence between the states $|S\alpha\rangle'$ and $|T_+\beta\rangle'$, let it evolve for half period and perform a field jump $B_{LC}^{(1)} \rightarrow B_0$. If the timing is properly set, the states $|S\alpha\rangle'$ and $|T_+\beta\rangle'$ exchange populations. In both cases, there is a single step of redistributing the populations. The resulting spin order is the same as in the case of the adiabatic field inversion. The experiments exploiting adiabatic passage are, most likely, easier to implement as they do not require precise control of the timing.

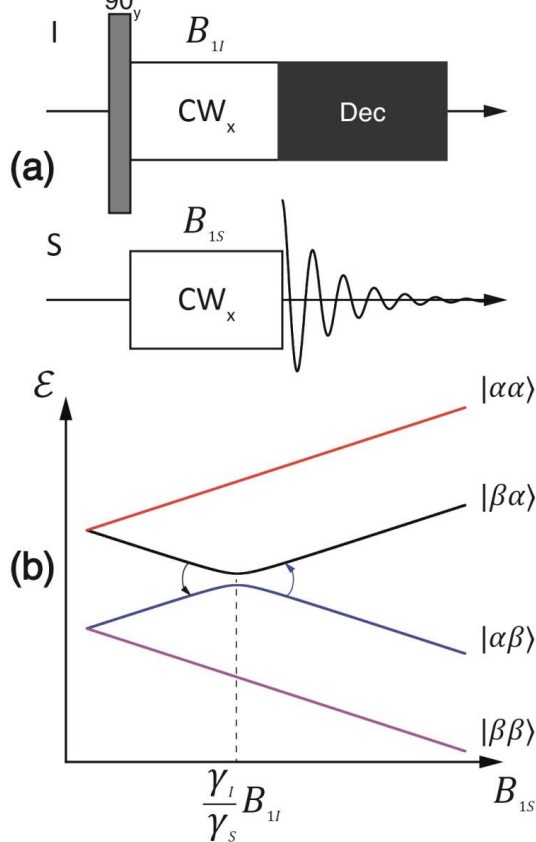

**Figure 6.** (a) experimental protocol for the cross polarization experiment (b) Adiabatic energy levels of the system in the doubly rotating frame. Simulation parameters: $I$ and $S$ nuclei are $^1$H and $^{13}$C respectively, $\omega_{1I}/2\pi = 25$ kHz, $H_{zz}/2\pi = 3$ kHz.



### 3.2 Cross-polarization

Cross-Polarization (CP) is a widely used method (Hartmann and Hahn, 1962; Pines et al., 1972; Hediger et al., 1994) to enhance NMR signals of "rare" nuclei in high-field NMR experiments, in particular, in solid-state NMR. The idea of CP is to transfer polarization from protons, hereafter denoted as $I$ spins, to "insensitive" nuclei, hereafter $S$ spins. Here we consider polarization transfer in a two-spin $IS$ system with $\gamma_I > \gamma_S$.

In the CP experiment (Hartmann and Hahn, 1962), see **Figure 6a**, the $I$ spins are first flipped by a $90°$ pulse, here a $90_y$ pulse, and then the transverse magnetization is locked by a continuous-wave (CW) pulse. After that, an RF-pulse is applied at the frequency of the $S$ spins. When the amplitudes of the two RF-fields are set in a proper way, see explanation below, the transverse polarization is transferred from the $I$ spins to $S$ spins. To detect this polarization, the RF-field applied to the $S$ spins is instantaneously turned off. Polarization transfer enables enhancement of the NMR signals of the $S$ spins due to the transfer of the higher polarization of the $I$ spins.

To describe this experiment, we write down the Hamiltonian in the doubly rotating frame

$$\widehat{\mathcal{H}}_{drf} = e^{i\omega_S t \hat{S}_z} e^{i\omega_I t \hat{I}_z} \widehat{\mathcal{H}} e^{-i\omega_I t \hat{I}_z} e^{-i\omega_S t \hat{S}_z} = \omega_{1I}\hat{I}_x + \omega_{1S}\hat{S}_x + \widehat{\mathcal{H}}_C \tag{32}$$

In such a frame the Zeeman interactions of the two spins are time-independent; for simplicity we assume that they are applied exactly on resonance so that the spins interact only with the RF-fields, here $\omega_{1I} = -\gamma_I B_{1I}$ and $\omega_{1S} = -\gamma_S B_{1S}$. The coupling term $\widehat{\mathcal{H}}_C$ is time-dependent and contains contributions, which oscillate at the frequencies $\omega_I$, $\omega_S$, $(\omega_I + \omega_S)$ and $(\omega_I - \omega_S)$. Such terms rapidly average out to zero; the only exception is given by the $zz$-term, $\widehat{\mathcal{H}}_{zz} = H_{zz}\hat{I}_z\hat{S}_z$, which commutes with $e^{i\omega_S t \hat{S}_z}$ and $e^{i\omega_I t \hat{I}_z}$ and remains time-independent in the doubly rotating frame. In the following, it is convenient to go to the doubly tilted frame, in which the quantization axes are parallel to the effective fields, i.e., to the $x$-axes of the doubly rotating frame. In the new frame, the Hamiltonian takes the form

$$\widehat{\mathcal{H}}_{drf} = \widehat{\mathcal{H}}_0 + \hat{V}, \tag{33}$$

$$\widehat{\mathcal{H}}_0 = \omega_{1I}\hat{I}_z + \omega_{1S}\hat{S}_z, \quad \hat{V} = H_{zz}\hat{I}_x\hat{S}_x = \frac{1}{4}H_{zz}\{\hat{I}_+\hat{S}_+ + \hat{I}_+\hat{S}_- + \hat{I}_-\hat{S}_+ + \hat{I}_-\hat{S}_-\}$$

These expressions are obtained from eq. (32) by making a substitution of spin operators: $\hat{I}_x, \hat{S}_x \to \hat{I}_z, \hat{S}_z$ and $\hat{I}_z\hat{S}_z \to \hat{I}_x\hat{S}_x$. The initial state of the spin system in the doubly tilted frame can be described by the following density matrix ($I$ spins are polarized along the corresponding RF-field)

$$\rho_i = \frac{1}{4}\hat{1} + M_I\hat{I}_z \tag{34}$$

The eigenstates of $\widehat{\mathcal{H}}_0$ are obviously the Zeeman states $|1\rangle = |\alpha\alpha\rangle$, $|2\rangle = |\alpha\beta\rangle$, $|3\rangle = |\beta\alpha\rangle$ and $|4\rangle = |\beta\beta\rangle$. In this basis the density matrix in eq. (34) can be written as:

$$|\rho) = \begin{pmatrix} \frac{1}{4} + \frac{M_I}{2} \\ \frac{1}{4} + \frac{M_I}{2} \\ \frac{1}{4} - \frac{M_I}{2} \\ \frac{1}{4} - \frac{M_I}{2} \end{pmatrix} \tag{35}$$

The perturbation term, which contains the raising and lowering spin operators, can mix the states $|1\rangle$ and $|4\rangle$ as well as $|2\rangle$ and $|3\rangle$. When $\omega_{1I}$ and $\omega_{1S}$ are of the same sign, the states $|2\rangle$ and $|3\rangle$ have a crossing, which can be turned into LAC by the $\hat{V}$ term, see **Figure 6b**. The LC condition





$$\omega_{1I} = \omega_{1S} \quad \Leftrightarrow \quad \gamma_I B_{1I} = \gamma_S B_{1S} \tag{36}$$

is known as the Hartmann-Hahn condition (Hartmann and Hahn, 1962). In accordance with this condition, the fields $B_{1I}$ and $B_{1S}$ should be set inverse proportional to the corresponding gyromagnetic ratios, i.e., $\frac{B_{1S}}{B_{1I}} = \frac{\gamma_I}{\gamma_S}$. By virtue of the perturbation term, the populations of the states $|2\rangle$ and $|3\rangle$ are redistributed and polarization transfer takes place. As a result, the density matrix takes the form

$$|\rho'\rangle = \widehat{\Pi}^{(\alpha\beta,\beta\alpha)}|\rho\rangle = |\rho\rangle = \begin{pmatrix} \frac{1}{4} + \frac{M_I}{2} \\ \frac{1}{4} + \frac{M_I}{2}(1 - 2\Delta) \\ \frac{1}{4} - \frac{M_I}{2}(1 - 2\Delta) \\ \frac{1}{4} - \frac{M_I}{2} \end{pmatrix} \tag{37}$$

Hence, in the ideal case $M_S' = (M_S|\rho') \xrightarrow{\Delta \to 1} M_I$ and $z$-polarization is completely transferred to the $S$ spin.
In the non-tilted rotating frame this would correspond to the transfer of transverse polarization among the spins of the heteronuclei.

The CP experiment can be done is a different way (Metz et al., 1994). The RF-field $B_{1S}$ can be increased in an adiabatic fashion from a value below $\frac{\gamma_I}{\gamma_S} B_{1I}$ (corresponding to the LC) to a value above this field, in order to enable passage through the LAC. The result of such an experiment, ramped-CP, will be
the same as for conventional CP: passage through the LAC will enable population swapping between the same states, $|2\rangle$ and $|3\rangle$. Such a technique is often more robust, as explained above.

**3.3 Singlet order**

Experiments with long-lived singlet order are drawing increased attention, as they allow one to investigate various slow processes and to preserve non-thermal spin order from relaxation losses (Levitt,
2012; Carravetta and Levitt, 2004; Carravetta et al., 2004). Presently, there is a number of NMR methods, reviewed in detail by Pileio (Pileio, 2017), known to convert magnetization into singlet order and to perform backward conversion of such a long-lived order into detectable magnetization. In strict terms, the long-lived order is given by the expectation value of the singlet order operator $\langle SO \rangle$. The singlet order operator is written as:

$$\widehat{SO} = |S\rangle\langle S| - \frac{1}{3}\left(|T_+\rangle\langle T_+| + |T_0\rangle\langle T_0| + |T_-\rangle\langle T_-|\right) \tag{38}$$

In the present work, we only focus on LAC-based methods, which can be applied to pairs of nearly-equivalent spins 1/2, meaning that the difference $\{\omega_a - \omega_b\}$ in their Zeeman interaction with the external field is much smaller than the spin-spin coupling strength $J$. In the weak coupling regime LAC-based consideration is typically not applicable, whereas in strongly coupled spin pairs the magnetization-to-singlet conversion is commonly occurring at LACs in the RF-rotation frame, carried out in the manner of
SLIC (Spin-Locking Induced Crossing).(DeVience et al., 2013)

The Hamiltonian of a homonuclear two-spin system, comprising spins $I_a$ and $I_b$, in the presence of an RF-field can be written as follows in the rotating frame:

$$\widehat{\mathcal{H}} = \delta\omega_a \hat{I}_{az} + \delta\omega_b \hat{I}_{bz} + \omega_1\{\hat{I}_{ax} + \hat{I}_{bx}\} + 2\pi J(\hat{\mathbf{I}}_a \cdot \hat{\mathbf{I}}_b) \tag{39}$$

Here $\delta\omega_{a,b} = \omega_{a,b} - \omega_{rf}$, where $\omega_{a,b}$ stand for the NMR frequency of the corresponding spin and $\omega_{rf}$ is the RF-frequency. The definition of the main term and the perturbation is then as follows:



$$\hat{\mathcal{H}}_0 = \langle\delta\omega\rangle\{\hat{I}_{az} + \hat{I}_{bz}\} + \omega_1\{\hat{I}_{ax} + \hat{I}_{bx}\} + 2\pi J(\hat{\mathbf{I}}_a \cdot \hat{\mathbf{I}}_b), \qquad \hat{V} = \omega_\Delta\{\hat{I}_{az} - \hat{I}_{bz}\} \tag{40}$$

where $\langle\delta\omega\rangle = \frac{1}{2}\{\delta\omega_a + \delta\omega_b\}$ and $\omega_\Delta = \frac{1}{2}\{\delta\omega_a - \delta\omega_b\} = \frac{1}{2}\{\omega_a - \omega_b\}$. Hence, the perturbation is given by the small difference in the resonance frequencies of the two spins. To determine the eigenvalues and eigenstates of the main Hamiltonian it is convenient to tilt the reference frame such that the new $z$-axis is parallel to the effective field vector $\boldsymbol{\omega} = (\omega_1, 0, \langle\delta\omega\rangle)$. In this frame the $\hat{\mathcal{H}}_0$ term takes the form:

$$\hat{\mathcal{H}}_0 = \omega_{eff}\{\hat{I}_{az} + \hat{I}_{bz}\} + 2\pi J(\hat{\mathbf{I}}_a \cdot \hat{\mathbf{I}}_b) \tag{41}$$

where $\omega_{eff} = \sqrt{\omega_1^2 + \langle\delta\omega\rangle^2}$. The scalar coupling term remains unchanged, since the operator $(\hat{\mathbf{I}}_a \cdot \hat{\mathbf{I}}_b)$
is invariant to spatial rotations. The eigenstates of $\hat{\mathcal{H}}_0$ correspond to the singlet-triplet basis of states in the tilted frame:

$$\mathbb{ST}_t = \hat{\Psi}_x(\theta_t)\{|T_+\rangle, |S\rangle, |T_0\rangle, |T_-\rangle\} = \{|T_+\rangle', |S\rangle, |T_0\rangle', |T_-\rangle'\} \tag{42}$$

Where $\theta_t = \tan^{-1}\omega_1/\delta\omega$. The primes in the notations of the triplet states indicate that they are defined in the tilted reference frame with $z||\boldsymbol{\omega}_{eff}$ (we do not use the prime for the $|S\rangle$ state, which is the same in any frame). The energies of these states of $\hat{\mathcal{H}}_0$ are:

$$\mathcal{E}_S = -\frac{3\pi}{2}J, \quad \mathcal{E}_{T_+} = \omega_{eff} + \frac{\pi}{2}J, \quad \mathcal{E}_{T_0} = \frac{\pi}{2}J, \quad \mathcal{E}_{T_-} = -\omega_{eff} + \frac{\pi}{2}J \tag{43}$$

The $\hat{\mathcal{H}}_0$ Hamiltonian has a single LC occurring when $\omega_{eff} = 2\pi|J|$, which is an $S$-$T_+$ or $S$-$T_-$ crossing (depending on the sign of $J$). The coupling term gives rise to mixing of the crossing states, hence, the LC is turned into a LAC. Let us now consider how spin mixing at this LAC can be exploited to perform spin order conversion.

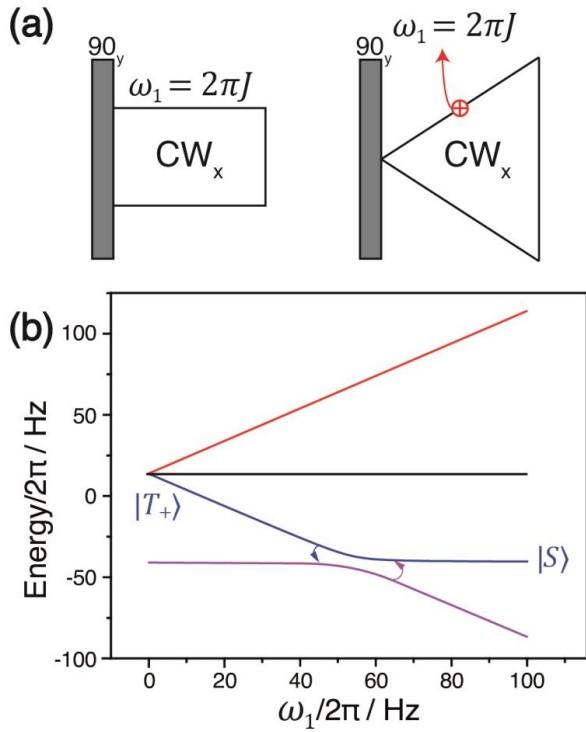

**Figure 7.** (a) The experimental protocol of SLIC (left) and adiabatic SLIC with linearly ramped RF-field amplitude (right). (b) Correlation diagram describing $T_+ \to S$ conversion at the LAC.



The simplest way to convert spin order is given by the SLIC (Spin-Locking Induced Crossing) (DeVience et al., 2013) method, which utilizes a resonant RF-pulse, i.e., $\langle \delta\omega \rangle = 0$, with $\omega_1 = 2\pi|J|$. Application of such a pulse brings the spin system to the LC, where the perturbation term becomes active. 470 Hence, $S\text{-}T_\pm$ mixing takes place, in the ideal case it swaps the populations of the two states, see **Figure 7**.

Efficient conversion of magnetization into singlet state requires that first the magnetization vector is set parallel to the effective field, in the case $\langle \delta\omega \rangle = 0$ the $\boldsymbol{\omega}_{eff}$ vector is parallel to the $x$-axis of the tilted frame. Hence, starting with $z$-polarization one should first apply a $90_y$ pulse and then apply a SLIC-pulse with the $x$-phase. Under such conditions the initial density matrix in the tilted frame takes the form

$$\rho_i = \frac{1}{4}\hat{1} + M_I\{\hat{I}_{az} + \hat{I}_{bz}\} \tag{44}$$

Represented as a state population vector, it is as follows

$$|\rho) = \begin{pmatrix} \frac{1}{4} + M_I \\ \frac{1}{4} \\ \frac{1}{4} \\ \frac{1}{4} - M_I \end{pmatrix} \tag{45}$$

Hence, the longitudinal magnetization in the tilted frame (corresponding to the transverse magnetization in the original frame) is non-zero, while the singlet order is zero, $\langle SO \rangle = 0$. By applying a SLIC-pulse, however, one can swap the populations of the states $|S\rangle$ and $|T_+\rangle'$. When the RF-field is resonant, i.e., $\delta\omega = 0$, and the tilt angle is $\theta_t = \pi/2$ we obtain that permutation occurs between the states $|S\rangle$ and 480 $|T_+\rangle'$, which is obtained from $|T_+\rangle$ after a $90_x$ rotation. Consequently, after spin mixing at the LAC the state populations become

$$|\rho') = \hat{\Pi}^{(S,T_+)}(\Delta = 1)|\rho) = \begin{pmatrix} \frac{1}{4} \\ \frac{1}{4} + M_I \\ \frac{1}{4} \\ \frac{1}{4} - M_I \end{pmatrix} \tag{46}$$

According to the definition given by eq. (37), the singlet order operator is as follows:

$$(SO| = \begin{pmatrix} -\frac{1}{3} & 1 & -\frac{1}{3} & -\frac{1}{3} \end{pmatrix} \tag{47}$$

Hence, we obtain that $\langle SO \rangle = (SO|\rho') = \frac{4}{3}M_I$ and the polarization is reduced. The same kind of pulse can be used to convert the singlet order back into transverse polarization.

A possible way (Theis et al., 2014a) to implement SLIC is to apply a pulse with time-dependent amplitude $\omega_1(t)$, which is varied in an adiabatic way such that the minimal $\omega_1$ is smaller than $2\pi|J|$ and the maximal $\omega_1$ is greater than $2\pi|J|$. In this particular case, it does not matter if $\omega_1$ is increased or decreased: the permutation of the populations is the same, namely, the $|S\rangle$ and $|T_+\rangle'$ populations are swapped. As in the previous example, spin order conversion by an adiabatic pulse is usually more robust, 490 although a pulse with $\omega_1 = 2\pi|J|$ provides faster conversion.

Spin order conversion by SLIC pulses is not the unique method of driving singlet-triplet transitions. It is also possible to apply off-resonant pulses to perform the desired conversion. At a first glance, by using



an RF pulse with $\langle \delta\omega \rangle \neq 0$ and with a ramped amplitude $\omega_1(t)$, designed such that the LC at $\omega_{eff} = 2\pi|J|$ is passed, one can perform the same kind of transformation as in the SLIC case. However, this is not true because the direction of $\mathbf{\omega}_{eff}$ changes upon variation of RF-field amplitude. Indeed, when $\omega_1 = 0$ the effective field is directed along the $z$-axis (for any small, but non-zero, value of $\omega_1$), since there is only the $\langle \delta\omega \rangle$-term in the Hamiltonian $\widehat{\mathcal{H}}_0$, whereas at $\omega_1 \gg \langle \delta\omega \rangle$ the effective field is parallel to the $x$-axis. As a consequence, a pulse with an adiabatically increased $\omega_1(t)$ converts the $z$-magnetization of spins into singlet order. A pulse with adiabatically decreased $\omega_1(t)$ converts the singlet order into $z$-magnetization. This type of conversion is exploited in the APSOC (Adiabatic Passage Spin Order Conversion) method (Pravdivtsev et al., 2016), which has an advantage that additional pulses are not required for locking spin magnetization; furthermore, there is no need to control the phase of the pulses.

### 3.4 Parahydrogen induced polarization

PHIP also frequently relying (Franzoni et al., 2013; Franzoni et al., 2012; Pravdivtsev et al., 2014b; Theis et al., 2014b; Pravdivtsev et al., 2013) on spin mixing occurring at LACs. In this section, we discuss possible methods for transferring PHIP to polarize "rare" spins, such as $^{13}C$ or $^{15}N$. We consider here a three-spin system, comprising two $I$ spins (protons), $I_a$ and $I_b$, prepared in the singlet state and an $S$ spin. Such a consideration is relevant in the context of transferring SABRE-derived polarization to rare spins, such as $^{15}N$. A number of methods has been suggested to solve this problem (Theis et al., 2014b; Theis et al., 2018; Knecht et al., 2018); here we provide a unified view on such methods. For simplicity, we assume that the $I$ spins are chemically equivalent, but not magnetically equivalent nuclei: the unequal $J_{IS}$ couplings lift the magnetic equivalence. We also consider a particular method of spin order transfer, assuming that it is performed at a high magnetic field by applying RF-excitation solely on the $S$ channel. In the rotating frame (with the frame rotation done only for the $I$ spins) the Hamiltonian of the spin system is as follows:

$$\widehat{\mathcal{H}} = \omega_H\{\hat{I}_{az} + \hat{I}_{bz}\} + \delta\omega_S\hat{S}_z + \omega_1\hat{S}_x + 2\pi J_{II}(\hat{\mathbf{I}}_a \cdot \hat{\mathbf{I}}_b) + 2\pi J_{IS}\hat{I}_{az}\hat{S}_x \tag{48}$$

Here $\omega_H$ is the proton NMR frequency, $\delta\omega_S$ is the offset of the RF-field from the frequency of the $S$ spins, $\omega_1$ is the RF-field strength expressed in the frequency units, $J_{II}$ is the couplings of the $I$ spins. For the $IS$ couplings we assume that there is interactions only for the $I_a$-$S$ spin pair and that $J_{IS} \ll J_{II}$ (so that perturbation theory treatment is applicable). For the same reason as explained above, in the $IS$ coupling term we keep only the products of $z$-operators. Hence, we set the main part of the Hamiltonian as

$$\widehat{\mathcal{H}}_0 = \omega_H\{\hat{I}_{az} + \hat{I}_{bz}\} + \delta\omega_S\hat{S}_z + \omega_1\hat{S}_x + J_{II}(\hat{\mathbf{I}}_a \cdot \hat{\mathbf{I}}_b) \tag{49}$$

and the perturbation as

$$\hat{V} = 2\pi J_{IS}\hat{I}_{az}\hat{S}_z \tag{50}$$

Now we again go to the tilted frame and modify the Hamiltonian in the following way for the main term

$$\widehat{\mathcal{H}}_0 = \omega_H\{\hat{I}_{az} + \hat{I}_{bz}\} + \omega_{S,eff}\hat{S}_z + 2\pi J_{II}(\hat{\mathbf{I}}_a \cdot \hat{\mathbf{I}}_b) \tag{51}$$

and for the perturbation term

$$\hat{V} = 2\pi J_{IS}\{\cos\theta_{eff}\,\hat{I}_{az}\hat{S}_z + \sin\theta_{eff}\,\hat{I}_{az}\hat{S}_x\} \tag{52}$$

Here $\omega_{S,eff} = \sqrt{\omega_1^2 + \delta\omega_S^2}$ and $\theta_{eff}$ is the tilt angle; $\tan\theta_{eff} = \frac{\omega_1}{\delta\omega_S}$. Here the frame tilt is introduced only for the $S$ spin, which is subject to RF-excitation.

The next step is solving the eigen-problem of the unperturbed Hamiltonian. To do so, we introduce a suitable basis, which is given by the direct product of the singlet-triplet bases of each spin pair: $\{|S\rangle, |T_+\rangle, |T_0\rangle, |T_-\rangle\}_{II} \otimes \{|\alpha'\rangle, |\beta'\rangle\}_S$, in the basis of the $S$ spin the primes indicate that the Zeeman states are written in the tilted frame. In this basis, the $\widehat{\mathcal{H}}_0$ Hamiltonian is diagonal. It is then



straightforward to evaluate the diabatic energy levels. One can determine that two LCs emerge, when the following matching conditions are fulfilled

$$\mathcal{E}_{S\alpha'} = \mathcal{E}_{T_0\beta'}, \qquad \omega_{S,eff} = 2\pi J_{II}$$
$$\mathcal{E}_{S\beta'} = \mathcal{E}_{T_0\alpha'}, \qquad \omega_{S,eff} = -2\pi J_{II}$$

(53)

Here we consider only LCs in the manifold of $|S\rangle$ and $|T_0\rangle$ states. The reason is that the $|S\rangle$ states and $|T_{\pm}\rangle$ are split by the large proton Zeeman interaction, $\omega_H$, and the corresponding crossings cannot occur at high magnetic fields; furthermore, there is no perturbation term, which would mix these states.

Therefore, in the present case of single-frequency excitation it is sufficient to consider only $S$-$T_0$ mixing of the $I$ spins.

At each of the two LCs, the perturbation terms become active: the $\hat{I}_{az}$ operator can mix the $|S\rangle$ and $|T_0\rangle$ states, while the $\hat{S}_x$ operator can mix the $|\alpha\rangle$ and $|\beta\rangle$ states. One should only be careful that when the matching condition

$$\omega_{S,eff} = \pm 2\pi J_{II}$$

(54)

is fulfilled the $\theta_{eff}$ should not be approaching zero (which is the case when $\delta\omega \approx 2\pi J_{II} \gg \omega_1$): under such conditions the coupling term becomes too small to provide fast and efficient exchange of the state populations at the LAC. Of course, both conditions $\omega_{S,eff} = \pm 2\pi J_{II}$ cannot be fulfilled simultaneously. Therefore, for the sake of clarity, we assume that $\omega_{S,eff} = 2\pi J_{II}$. The relevant energy levels are shown in **Figure 8**.

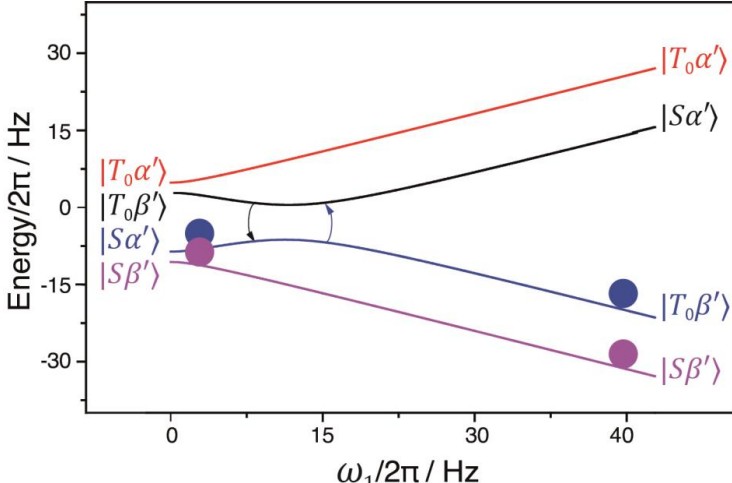


**Figure 8.** Population swapping upon increase of the amplitude of the RF-field applied to the $S$ spin. Simulation parameters: $I$ and $S$ nuclei are $^1$H and $^{13}$C respectively, $J_{II} = 11.5$ Hz, $J_{IS} = 13.7$ Hz, $\delta\omega = 2$ Hz. See text for further detail.

Now, let us consider the LAC-driven spin-dynamics of the process. In the case $J_{II} \gg J_{IS}$ the initial density matrix can be written as:

$$|\rho\rangle \approx \frac{1}{2}|S\alpha'\rangle + \frac{1}{2}|S\beta'\rangle$$

(55)

After population swapping the density matrix becomes

$$|\rho\rangle' = \hat{\Pi}^{(S\alpha, T_0\beta)}|\rho\rangle = \frac{1}{2}|T_0\beta'\rangle + \frac{1}{2}|S\beta'\rangle$$

(56)





That is, spin mixing gives rise to population exchange between the states $|S\alpha'\rangle$ and $|T_0\beta'\rangle$. As a consequence, singlet order is converted into magnetization of the $S$ spin. The resulting polarization of the $S$ spin is then as follows:

$$M_S' = (S_z'|\rho)' \approx \frac{1}{2}(S_z'|T_0\beta') + \frac{1}{2}(S_z'|S\beta') = \frac{1}{2} \tag{57}$$

Here $M_S'$ is the magnetization value in the tilted frame. The resulting spin order of the $S$ spins depends on how the experiment is carried out. In the simplest case, where a single pulse with $\omega_{S,eff} = 2\pi J_{II}$ is applied for a sufficiently long time (so that spin mixing can occur) the $S$ spin is polarized along the $\boldsymbol{\omega}_{S,eff}$ vector. In the situation $\delta\omega = 0$ and $\omega_1 = 2\pi J_{II}$ (resonant pulse) magnetization of the $S$ spin is the purely transverse magnetization: the $\hat{S}_z$ spin order in the tilted frame corresponds to $\hat{S}_x$ in the non-tilted frame. If it is necessary to generate longitudinal magnetization, an additional RF-pulse should be applied (Theis et al., 2014b). By applying a pulse with $\omega_{S,eff} = 2\pi J_{II}$ and $\delta\omega \neq 0$, one can again generate the $\hat{S}_z'$ order (Knecht et al., 2018) in the tilted frame, which corresponds to spin order

$$\hat{S}_z' = \hat{S}_z \cos\theta_{eff} + \hat{S}_x \sin\theta_{eff} \tag{58}$$

in the non-tilted frame. Hence, the magnetization vector has transverse as well as longitudinal components. If the RF-pulse is applied such (Theis et al., 2018) that $\omega_1(t)$ is adiabatically reduced to zero in such a way that the LC $\omega_{S,eff} = 2\pi J_{II}$ is passed and $\delta\omega \neq 0$ the $S$ spin is polarized along the $\boldsymbol{\omega}_{S,eff}$ vector, which becomes parallel to $z$-when $\omega_1$ becomes zero. It means that longitudinal magnetization of the $S$ spins is generated. It is important that necessarily $\delta\omega \neq 0$ in this case: when the offset from the resonance frequency is zero, the effective field does not have any preferred direction and the $S$ spins cannot be preferentially polarized parallel or anti-parallel to the external magnetic field.

A similar situation arises upon transfer of the singlet order into the magnetization of heteronuclei in a four-spin system of the AA′XX′ type. In this situation, the Hamiltonian is written as follows (in the RF-rotating frame for the $S$ spins):

$$\hat{\mathcal{H}}_0 = \omega_H\{\hat{I}_{az} + \hat{I}_{bz}\} + \delta\omega_S(\hat{S}_{az} + \hat{S}_{bz}) + \omega_1(\hat{S}_{ax} + \hat{S}_{bx}) + 2\pi J_{II}(\hat{\mathbf{I}}_a \cdot \hat{\mathbf{I}}_b)$$
$$+ 2\pi J_{SS}(\hat{\mathbf{S}}_a \cdot \hat{\mathbf{S}}_b) \tag{59}$$

The perturbation term cis given by expression

$$\hat{V} = 2\pi J_{IS}\{\hat{I}_{az}\hat{S}_{az} + \hat{I}_{bz}\hat{S}_{bz}\} \tag{60}$$

Now it is convenient to rewrite the Hamiltonians in the tilted frame. The main Hamiltonian becomes

$$\hat{\mathcal{H}}_0 = \omega_H\{\hat{I}_{az} + \hat{I}_{bz}\} + \omega_{S,eff}(\hat{S}_{az} + \hat{S}_{bz}) + 2\pi J_{II}(\hat{\mathbf{I}}_a \cdot \hat{\mathbf{I}}_b) + 2\pi J_{SS}(\hat{\mathbf{S}}_a \cdot \hat{\mathbf{S}}_b) \tag{61}$$

and the perturbation is

$$\hat{V} = 2\pi J_{IS}\{\cos\theta_{eff}(\hat{I}_{az}\hat{S}_{az} + \hat{I}_{bz}\hat{S}_{bz}) + \sin\theta_{eff}(\hat{I}_{az}\hat{S}_{ax} + \hat{I}_{bz}\hat{S}_{bx})\} \tag{62}$$

Here $\omega_{S,eff} = \sqrt{\omega_1^2 + \delta\omega_S^2}$ and $\theta_{eff}$ is the tilt angle; $\tan\theta_{eff} = \frac{\omega_1}{\delta\omega_S}$. The eigen-basis for the $\hat{\mathcal{H}}_0$ Hamiltonian for now is given by the direct product of the singlet-triplet bases in each spin pair: $\{|S\rangle, |T_+\rangle, |T_0\rangle, |T_-\rangle\}_{II} \otimes \{|S\rangle, |T_+'\rangle, |T_0'\rangle, |T_-'\rangle\}_{SS}$, in the basis of the $S$ spins the primes indicate that the singlet-triplet states are written in the tilted frame (there is no frame tilt introduced for the $I$ spins).

The perturbation term (61) can drive $S \rightarrow T_0$ transitions for the $I$ spins accompanied by $S \rightarrow T_\pm'$ transitions for the $S$ spins resulting in the increasing the $|T_\pm'\rangle$ populations and, consequently, causing the enhanced heteronuclei magnetization along the RF-field directions. The LCs of the system are the following (LC conditions are also specified):

$$\mathcal{E}_{SS} = \mathcal{E}_{T_0T_+'}, \qquad \omega_{S,eff} = -2\pi(J_{II} + J_{SS}) \tag{63}$$



$$\mathcal{E}_{SS} = \mathcal{E}_{T_0 T'_-}, \qquad \omega_{S,eff} = 2\pi(J_{II} + J_{SS})$$

The generalized LACs condition is then $\omega_{S,eff} = \pm 2\pi(J_{II} + J_{SS})$. The spin dynamics and polarization behavior is similar to the three-spin case described above; hence, we do not consider further detail here. In order to learn more about this subject, the reader is advised to read previous publications (Knecht et al., 2018; Theis et al., 2014b).

Finally in this section, we would like to note that similar LAC-driven spin dynamics have been reported for homonuclear systems of the AA′MM′ type, where AA′ and MM′ stand for the two groups of chemically equivalent but magnetically non-equivalent spins, with the AA′ spins prepared in the singlet state. Discussion of this case is beyond the scope of the present work. We only mention that spin order transfer is based on the same principles as those described above: upon RF-excitation polarization transfer occurs at LACs (in the rotating frame) and gives rise to polarization of the AA′ and MM′ spins along their respective effective fields. One can also vary the actual spin magnetization by introducing a single RF-pulse, which brings the spin system to an LAC, or by passing through LACs using adiabatically ramped RF-field amplitudes. Further information can be found in the original publications (Pravdivtsev et al., 2014b; Franzoni et al., 2013).

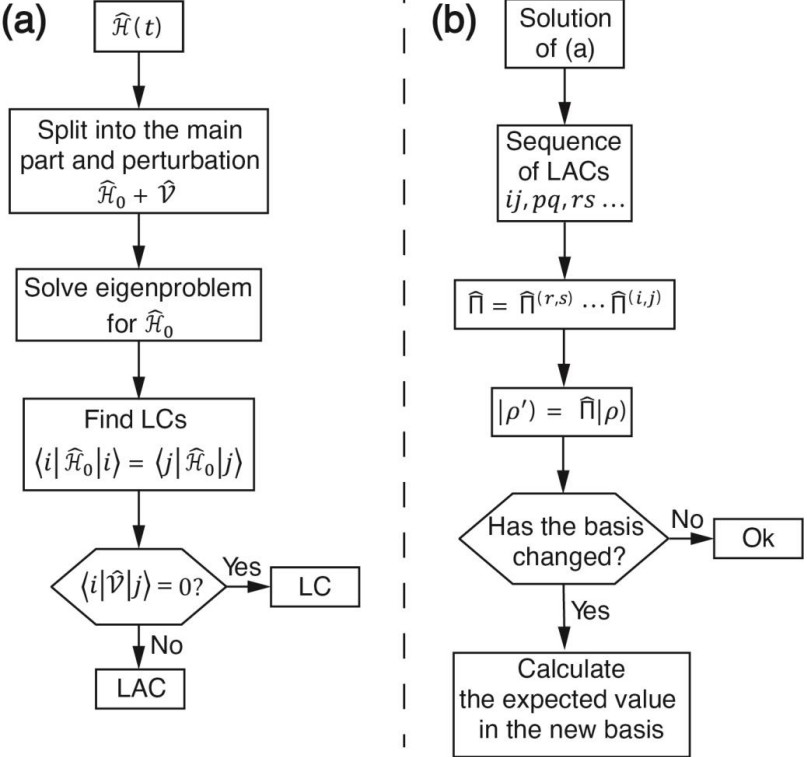

**Figure 9.** Flow-chart diagram indicating (a) the way to find all LACs and (b) to calculate the spin dynamics due to LACs.

## 4 Conclusions and Outlook

In this work, we present a general approach to treat spin mixing occurring at LACs. The approach is formulated assuming that the spin system has a set of LACs, which do not overlap with each other, for the state described in terms of the populations of diabatic states, i.e., we ignore the possible presence of spin coherences in the initial and final state. Upon variation of a control parameter (magnetic field strength, RF-frequency, RF-field strength) the spin system passes through LACs and permutations of the



state populations occur. Introducing the operators of permutations, we can compute the final spin order. We also take into account that upon variation of the control parameter the basis of the diabatic eigenstates may be altered. This consideration of the spin dynamics proposed here is summarized by a flow-chart diagram, shown in **Figure 9**.

The treatment presented here is supported by a number of examples. These examples are dealing
with spin order conversion via adiabatic passage through zero field, with cross-polarization, with singlet-state NMR and with PHIP. To conclude, utilizing LACs provides powerful methods to manipulate spin order and to design experimental protocols for robust and efficient spin order conversion. LAC based methods have proven to be a useful tool. For instance, in our lab we have developed several methods based on harnessing LACs, such as the APSOC method and techniques for manipulating PHIP.

**Acknowledgements**

We acknowledge Prof. J. Matysik for inspiring us to describe the representation of LAC driven spin mixing, to Prof. G. Bodenhausen for valuable comments on the text of the manuscript and Prof. M. H. Levitt for drawing our attention to the discussion of the "basis rotation" concept. This work has been supported by the Russian Foundation for Basic Research (grant No. 20-53-15004) and the Ministry of
Science and Education of the Russian Federation.

**Appendix A. Degenerate perturbation theory**

Here we present calculations of the effective coupling element $\mathcal{V}_{kl}$ in the situation where $|\psi_k\rangle$ and $|\psi_l\rangle$ are not mixed by the perturbation, but are mixed with other states $|\psi_m\rangle$. If we assume that there is only one such state, we can evaluate the second-order correction to the wavefunctions:

$$\left|\psi_k^{(2)}\right\rangle \approx |\psi_k\rangle + \frac{\mathcal{V}_{mk}}{\mathcal{E}_k^0 - \mathcal{E}_m^0}|\psi_m\rangle, \qquad \left|\psi_l^{(2)}\right\rangle \approx |\psi_l\rangle + \frac{\mathcal{V}_{ml}}{\mathcal{E}_l^0 - \mathcal{E}_m^0}|\psi_m\rangle \qquad \text{(A1)}$$

The new wavefunctions can be mixed, because $\mathcal{V}_{km}$ and $\mathcal{V}_{lm}$ are both non-zero. Assuming $\mathcal{E}_k^0 \approx \mathcal{E}_l^0$ we can estimate the matrix element, which mixes them, as follows

$$\mathcal{V}_{kl}^{eff} \approx \frac{\mathcal{V}_{km}\mathcal{V}_{ml}}{\mathcal{E}_k^0 - \mathcal{E}_m^0} \qquad \text{(A2)}$$

If there are multiple states $|\psi_m\rangle$, through which the $|\psi_k\rangle$ and $|\psi_l\rangle$ are coupled, we generalize this expression as

$$\mathcal{V}_{kl}^{eff} \approx \sum_{m \neq k,l} \frac{\mathcal{V}_{km}\mathcal{V}_{ml}}{\mathcal{E}_k^0 - \mathcal{E}_m^0} \qquad \text{(A3)}$$

The effective coupling element vanished only when for any $m$ both $\mathcal{V}_{km}$ and $\mathcal{V}_{lm}$ are zero, meaning that
$|\psi_k\rangle$ and $|\psi_l\rangle$ must belong to different blocks of a block-diagonal Hamiltonian. In this situation, a true LC between these states is possible; otherwise, it is converted into a LAC.

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
