# Peer review of "Representation of population exchange at level anti-crossings"

_Magnetic Resonance, 2020_

## Referee Comment (RC1) · Anonymous Referee #1 · 2 Oct 2020

In this work, Bodin and Ivanov give a thorough description of the use of the level anti-crossing concept to analyse population exchange in NMR experiments. The basic physical principles are given, and several applications are analysed with this formalism. This article is particularly welcome, considering the number and the importance of recent studies that rely on the LAC description to design new experiments. It is well written, in a clear and pedagogical way. I recommend publication in Magnetic Resonance, optionnally after the following comments have been addressed.

Main comments:

The formalism described here is very powerful to identify the possibility of population exchange (existence of a LAC) and qualitatively what happens to the population. In the examples analysed here, no expression is given for the parameter Delta, as the

function of exprimental parameters, be it for the adiabatic or the non adiabatic transfers. In other words, Eq. 7 and 8 are not reused. Is the formalism moslty useful for qualitative analyses ?

The notion that a LC is "converted into a LAC" is sometimes confusing, especially when the conversion seems to be described as a dynamic process while the perturbation is an internal interaction of the spin system. Is that common QM jargon ? For example, on l. 302 "which are never converted into a LAC" is confusing, since nothing can change anyway. Perhaps "which are not converted into a LAC by the perturbation" ?

The authors could explicitly state whether nuclear spin relaxation can also be described with this approach. Also, for adiabatic passage "the population adjust to the slow varation of the adiabatic eigenstates"; is that a coherent process ?

It would be helpful to explain in each case the basis chosen to write the initial density matrix. For example, in Eq. 20 polarisation operator are used, while in Eq. 29 populatin operators are used. Also the quantities $M_I$ and $M_S$ in Eq. 20 should be better defined. How to they relate to the populations ? What is their bounds ?

For the CP examples, the operators to move to the tilted frame, as well as the coupling terms, could be given explicitly.

Additional comments:

l. 64: the sentence "Such symmetry breaking can only occur under special conditions, which correspond to LACs" is a bit mysterious

l. 271: "Examples, in which basis rotation is taking place, are discussed below." Please clarify which examples

l. 491-510 are difficult to follow without a figure and/or a more detailed description.

---

## Referee Comment (RC2) · Anonymous Referee #2 · 1 Nov 2020

This manuscript elegantly treat the LAC phenomena in magnetic resonance and outlines several cases where these could be of use. This will be an extremely useful article for the understanding of LAC and related aspects and making use of them in fields like PHIP and/or singlet state NMR.

I have a few minor comments:

*The ket notation looks like a curved bracket in many of the equations. This is confusing.

*Can the authors justify the form of the \Pi operator in Eq. 14?

*The use of superoperators is also a bit confusing. For instance, in Eq. 18, the superoperator acts on state kets and not operators.

[Figure]

*Can the authors say in which cases the basis rotation is equivalent to a physical rotation of spins in the 3D space.

*Can MAS be included in the arguments given here?

I recommend publication of this manuscript with minor revisions, as noted above, if required.

―――――――――――――――――――

---

## Author Comment (AC1) · 15 Nov 2020

We are thankful to reviewers for useful comments, which have helped us to improve the paper. A detailed point-by-point response to their questions is given below.

To this submission, we also attach a highlighted revision file, in which all changes are marked in color.

**Referee #1**

In this work, Bodin and Ivanov give a thorough description of the use of the level anticrossing concept to analyse population exchange in NMR experiments. The basic physical principles are given, and several applications are analysed with this formalism. This article is particularly welcome, considering the number and the importance of recent studies that rely on the LAC description to design new experiments. It is well written, in a clear and pedagogical way. I recommend publication in Magnetic Resonance, optionally after the following comments have been addressed.

Main comments:

The formalism described here is very powerful to identify the possibility of population exchange (existence of a LAC) and qualitatively what happens to the population. In the examples analysed here, no expression is given for the parameter Delta, as the function of exprimental parameters, be it for the adiabatic or the non adiabatic transfers. In other words, Eq. 7 and 8 are not reused. Is the formalism moslty useful for qualitative analyses ?

*Response*: eqs. 7 and 8 were indeed not reused in the original version. Nonetheless, we would like to keep them so that a reader would be able to use our approach in cases, where population exchange is incomplete. In most examples considered here, we derive the final expressions, valid also in the case of incomplete population exchange. In fact, eqs. (7) and (8) make the approach quantitative. We have added some comments, clarifying how experimental parameters should be set, using eqs. (7) and (8).

The notion that a LC is "converted into a LAC" is sometimes confusing, especially when the conversion seems to be described as a dynamic process while the perturbation is an internal interaction of the spin system. Is that common QM jargon? For example, on l. 302 "which are never converted into a LAC" is confusing, since nothing can change anyway. Perhaps "which are not converted into a LAC by the perturbation" ?

**Response**: The "LC conversion into a LAC" is to some extent a dynamical process (in order to distinguish LC from LAC one needs time of about $1/=\mathcal{V}_{kl}$), but here we rather consider it as an intrinsic property of the spin system. Such a modification of the energy levels is taking place only when there is a perturbation present, which can mix the crossing levels. Whether it exist or not, depends on symmetry properties of a spin system. This is, perhaps, not a common QM jargon and we agree that such notion may be confusing. Thus we change "is converted into" to "is turned into".

In the example from l.302 the perturbation may can be altered by using external fields that can turn LC into LACs. Nevertheless, the above perturbation can't do it, and we indicated it as "which are not converted into a LAC by the perturbation of Hamiltonian (19)" (highlighted by blue).

The authors could explicitly state whether nuclear spin relaxation can also be described with this approach. Also, for adiabatic passage "the population adjust to the slow varation of the adiabatic eigenstates"; is that a coherent process ?

**Response**: Nuclear relaxation cannot be described by this approach, at least, without a modification. It is primarily for the coherent processes. To point in out, we change the l.70 to "This contribution aims at a simple description of LAC-based **coherent** phenomena", which is highlighted by blue. The population adjustment is also the coherent process. However, in

conclusions we state that potentially one can extend the approach to treat relaxation effects as well, by introducing a relaxation superoperator.

It would be helpful to explain in each case the basis chosen to write the initial density matrix. For example, in Eq. 20 polarisation operator are used, while in Eq. 29 population operators are used.
**Response**: We tried to introduce clearly the basis kets every time before writing the equations, as we did before equation (20). Furthermore, we tried introduce the density matrix in both possible ways, e.g., compare (20 and (21). We also hope that our notations allow to avoid confusion: population operator uses state-space brackets $|P)$, while the other operators uses conventional hats.

Also the quantities M_I and M_S in Eq. 20 should be better defined. How to they relate to the populations ? What is their bounds ?
**Response**: We have added after Eq. (20) "*They also can be defined explicitly from population as $M_I = p_{\alpha\alpha} + p_{\alpha\beta} - p_{\beta\alpha} - p_{\beta\beta}$ and $M_S = p_{\alpha\alpha} - p_{\alpha\beta} + p_{\beta\alpha} - p_{\beta\beta}$ ranging from $1$ to $-1$*".

For the CP examples, the operators to move to the tilted frame, as well as the coupling terms, could be given explicitely.
**Response**: We have added a clarifying statement, discussing the frame rotation in terms of rotation operators. $\hat{\mathcal{H}}_0$ is given in Eq. (33), and the perturbation is given in the text below.

Additional comments:
l. 64: the sentence "Such symmetry breaking can only occur under special conditions, which correspond to LACs" is a bit mysterious
**Response**: This is explained now. We argue that this is usually a minor effect, which becomes important only at level crossings.
l. 271: "Examples, in which basis rotation is taking place, are discussed below." Please clarify which examples
**Response**: Now it is clarified. Such examples are discussed in section 3.4.
l. 491-510 are difficult to follow without a figure and/or a more detailed description
**Response**: We are unable to cover all possible examples in full detail (this would probably double the length of the paper). This case is thus only mentioned here. An interested reader can go to the original publication.

**Referee #2**
This manuscript elegantly treat the LAC phenomena in magnetic resonance and outlines several cases where these could be of use. This will be an extremely useful article for the understanding of LAC and related aspects and making use of them in fields like PHIP and/or singlet state NMR.

I have a few minor comments:
*The ket notation looks like a curved bracket in many of the equations. This is confusing.
**Response**: The curved brackets do not correspond to the ket notation for the wavefunction, this is the population operator. We deliberately use curved brackets for populations and density matrices, not angular brackets commonly used for wavefunctions. This is explicitly states after equation (11) that $|\psi_m) = |\psi_m\rangle\langle\psi_m|$. We put a clarifying remark to warn the reader after eq. 11.

*Can the authors justify the form of the \Pi operator in Eq. 14?
Response: This equation is briefly explained as follows.
It is easily seen, that this operator doesn't change the states $m \neq l, k$. When acting one the state, for example, $|\psi_k^i)$, one can write:
$$\hat{\Pi}_{mn}^{(kl)}|\psi_k^i) = (1 - \Delta)|\psi_k^f)(\psi_k^i|\psi_k^i) + \Delta|\psi_l^f)(\psi_k^f|\psi_k^i) = (1 - \Delta)|\psi_k^f) + \Delta|\psi_l^f)$$

This corresponds to the (13) equation. The situation for $|\psi_l^i\rangle$ is symmetrical.

*The use of superoperators is also a bit confusing. For instance, in Eq. 18, the superoperator acts on state kets and not operators.
**Response**: The superoperator acts on an operator (state operator with curly brackets). We hope that our quick remark (the answer to question 1) will eliminate the confusion.

*Can the authors say in which cases the basis rotation is equivalent to a physical rotation of spins in the 3D space.
**Response**: A clarifying statement is added: If the operator generating the basis rotation is $(\mathbf{n}, \hat{\mathbf{I}})$, the basis rotation coincides with the physical spin rotation. In the general case, this is no true.

*Can MAS be included in the arguments given here?
**Response**: This is true, MAS-NMR can be a good application. We have mentioned this in conclusions and added a few refs.